# Multi-ancestry meta-analysis of host genetic susceptibility to tuberculosis identifies shared genetic architecture

Haiko Schurz[1]*[†], Vivek Naranbhai[2,3,4,5,6†], Tom A Yates[7], James J Gilchrist[2,8], Tom Parks[2,9], Peter J Dodd[10], Marlo Möller[1], Eileen G Hoal[1], Andrew P Morris[11], Adrian VS Hill[2,12], International Tuberculosis Host Genetics Consortium

[1]DSI-NRF Centre of Excellence for Biomedical Tuberculosis Research, South African Medical Research Council Centre for Tuberculosis Research, Division of Molecular Biology and Human Genetics, Faculty of Medicine and Health Sciences, Stellenbosch University, Cape Town, South Africa; [2]Wellcome Centre for Human Genetics, University of Oxford, Oxford, United Kingdom; [3]Massachusetts General Hospital, Boston, United States; [4]Dana-Farber Cancer Institute, Boston, United States; [5]Centre for the AIDS Programme of Research in South Africa, Durban, South Africa; [6]Harvard Medical School, Boston, United States; [7]Division of Infection and Immunity, Faculty of Medical Sciences, University College London, London, United Kingdom; [8]Department of Paediatrics, University of Oxford, Oxford, United Kingdom; [9]Department of Infectious Diseases Imperial College London, London, United Kingdom; [10]School of Health and Related Research, University of Sheffield, Sheffield, United Kingdom; [11]Centre for Genetics and Genomics Versus Arthritis, Centre for Musculoskeletal Research, The University of Manchester, Manchester, United Kingdom; [12]Jenner Institute, University of Oxford, Oxford, United Kingdom

*For correspondence:
haikoschurz@gmail.com

[†]co-first authors

Group author details:
International Tuberculosis Host Genetics Consortium See page 16

Competing interest: The authors declare that no competing interests exist.

**Abstract** The heritability of susceptibility to tuberculosis (TB) disease has been well recognized. Over 100 genes have been studied as candidates for TB susceptibility, and several variants were identified by genome-wide association studies (GWAS), but few replicate. We established the International Tuberculosis Host Genetics Consortium to perform a multi-ancestry meta-analysis of GWAS, including 14,153 cases and 19,536 controls of African, Asian, and European ancestry. Our analyses demonstrate a substantial degree of heritability (pooled polygenic $h^2$ = 26.3%, 95% CI 23.7–29.0%) for susceptibility to TB that is shared across ancestries, highlighting an important host genetic influence on disease. We identified one global host genetic correlate for TB at genome-wide significance ($p<5 \times 10^{-8}$) in the human leukocyte antigen (HLA)-II region (rs28383206, p-value=$5.2 \times 10^{-9}$) but failed to replicate variants previously associated with TB susceptibility. These data demonstrate the complex shared genetic architecture of susceptibility to TB and the importance of large-scale GWAS analysis across multiple ancestries experiencing different levels of infection pressure.

## Editor's evaluation

This article describes an important multi-ancestry meta-analysis of genome-wide association studies of susceptibility to tuberculosis. It demonstrates substantial heritability from common genetic variants, although this varies across studies. The main finding of the article is a variant in the HLA region that affects tuberculosis risk, for which the evidence is solid. The results and methods will be of interest to infectious disease researchers and human genetics researchers. The article highlights

both the promise and challenges of performing multi-ancestry genetic association studies of infectious disease risk.

## Introduction

Tuberculosis (TB), caused by *Mycobacterium tuberculosis* (*Mtb*) and related species, remains a leading cause of death globally. Around one-quarter of the global population is estimated to show immunological evidence of prior exposure to *Mtb* (*Houben and Dodd, 2016*), and in 2019 an estimated 10 million people developed the disease, resulting in 1.4 million deaths (*WHO, 2020*). This disease burden could be substantially reduced with action to address the social determinants of disease and equitable scale-up of existing interventions. However, tools to prevent, diagnose, and treat TB could be improved if a better understanding of the underpinning pathophysiology could help identify those at greatest risk of the disease.

The role of host genetic factors in TB susceptibility has long been of significant interest. Over 100 candidate genes have been studied, but few associations have proven reproducible (*Naranbhai, 2016*). This failure to replicate may be a result of the modest size of many TB genome-wide association studies (GWAS), variability in phenotyping between studies, the impact of population-specific effects, the challenge of complex population structure in some high-burden settings (e.g., admixed individuals), and, possibly, pathogen variation (*Correa-Macedo et al., 2019*; *Daya et al., 2014a*; *Luo et al., 2019*; *Möller and Kinnear, 2020*; *Müller et al., 2021*; *Omae et al., 2017*; *Schurz et al., 2018*). Seventeen GWAS have been reported but only two loci replicate between studies (*Daya et al., 2014a*; *Schurz et al., 2018*; *Chimusa et al., 2014*; *The Wellcome Trust Case Control Consortium, 2007*; *Curtis et al., 2015*; *Mahasirimongkol et al., 2012*; *Qi et al., 2017*; *Thye et al., 2010*; *Thye et al., 2012*; *Quistrebert et al., 2021*; *Sveinbjornsson et al., 2016*; *Hong et al., 2017*; *Li et al., 2021*; *Luo et al., 2019*; *Zheng et al., 2018*; *Grant et al., 2016*; *Png et al., 2012*). The *WT1* locus, identified in cohorts from Ghana and Gambia, replicated in South Africa and Russia. The *ASAP1* locus identified in Russia was replicated through reanalysis of prior studies (*Correa-Macedo et al., 2019*; *Möller and Kinnear, 2020*).

To address these challenges, we established the International Tuberculosis Host Genetics Consortium (ITHGC) to study the host genetics of disease through collaborative and equitable data sharing (*Naranbhai, 2016*). The ITHGC includes 12 case–control GWAS from nine countries in Europe, Africa, and Asia (total of 14,153 pulmonary TB cases and 19,536 healthy controls). Inclusion of multiple ancestral groups in a multi-ancestry meta-analysis has the advantage of maximizing power and enhancing fine-mapping resolution to identifying true global associated variants that influence TB susceptibility across population groups.

Here we present the first analyses of the ITHGC dataset exploring host genetic correlates of TB susceptibility using a multi-ancestry meta-analysis approach, including fine-mapping of human leukocyte antigen (HLA) loci and estimation of genetic heritability.

## Results
### Study overview

In total, 12 GWAS from three major ancestral groups (European, African, and Asian) were included in this study (*Table 1*; a more detailed table outlining the selection of cases and controls is provided in *Supplementary file 1a*). All individual datasets were imputed and aligned to the same reference allele before association testing, using an additive genetic model, to obtain odds ratios (OR) and p-values to be used in the meta-analysis. For each individual study (for which we had raw genotyping data), the polygenic heritability was estimated, and HLA alleles were imputed for fine-mapping of the HLA regions.

The summary statistics from the individual GWAS of each dataset were used to conduct a combined, multi-ancestry meta-analysis using MR-MEGA and ancestry-specific (European, African, and Asian) fixed effects (FE) meta-analyses using GWAMA. Finally, the impact of infection pressure on the multi-ancestry meta-regression was assessed and the concordance in direction of effect for the reference allele between studies was investigated.

**Table 1.** Summary of ITHGC TB-GWAS datasets.

| Dataset | Population | Cases/ controls | TB prevalence per 100 ,000 pa | Estimated proportion of controls ever exposed to *Mtb* (±SD)* | #SNPs | Genotyping platform | Reference |
|---|---|---|---|---|---|---|---|
| China 1† | Asian | 483/587 | 89 | 0.302 (0.101) | 7,710,153 | Affymetrix Genome-Wide Human SNP Array 6.0 | thye@bni-hamburg.de (unpublished) |
| China 2† | Asian | 1290/1145 | 89 | 0.302 (0.101) | 9,769, 029 | Illumina Human OmniZhonghua-8 chips | magdakellis@gmail.com (unpublished) |
| China 3 | Asian | 972/1537 | 89 | 0.302 (0.101) | 9,726,450 | Illumina Human OmniZhonghua-8 chips | *Qi et al., 2017* |
| Thailand | Asian | 433/295 | 236 | 0.404 (0.112) | 6,723,358 | Illumina Human610-Quad | *Mahasirimongkol et al., 2012* |
| Japan | Asian | 751/3199 | 23 | 0.142 (0.125) | 9,051,051 | Illumina HumanHap550 | *Mahasirimongkol et al., 2012* |
| Russia† | European | 5914/6022 | 109 | 0.191 (0.093) | 10,878,777 | Affymetrix Genome-Wide Human SNP Array 6.0 | *Curtis et al., 2015* |
| Estonia | European | 239/7047 | 13 | 0.116 (0.093) | 10,611,556 | Illumina 370K | andres.metspalu@ut.ee (unpublished) |
| Germany† | European | 586/333 | 7.8 | 0.067 (0.081) | 10,602,193 | Illumina Omni2.5+exome | thye@bni-hamburg.de (unpublished) |
| Gambia† | African | 1316/1382 | 126 | 0.280 (0.089) | 18,634,017 | Affymetrix GeneChip 500K | *The Wellcome Trust Case Control Consortium, 2007* |
| Ghana† | African | 1359/1952 | 282 | 0.539 (0.198) | 19,029,214 | Affymetrix Genome-Wide Human SNP Array 6.0 | *Thye et al., 2010* |
| RSA(A)† ‡ | African | 19/577 | 717 | 0.436 (0.127) | 9,227,330 | Affymetrix 500k | *Daya et al., 2014b* |
| RSA(M)†‡ | African | 410/405 | 717 | 0.436 (0.127) | 11,371,838 | Illumina MEGA array | *Schurz et al., 2018* |

GWAS, genome-wide association studies; ITHGC, International Tuberculosis Host Genetics Consortium; *Mtb*, *Mycobacterium tuberculosis*; TB, tuberculosis.

*Estimated proportion of control individuals ever infected with *Mtb* by age 35–44 in 2010, based on data from Houben & Dodd.

†Raw genotyping data available.

‡RSA(A/M): South African admixed population (RSA) Affymetrix (A) and MEGA (M) array data.

## Polygenic heritability estimates suggest a genetic contribution to TB disease susceptibility

Twin studies estimate the narrow-sense heritability of susceptibility to TB at up to 80% (*Diehl and Von, 1936*; *Kallmann and Reisner, 1943*; *Comstock, 1978*), but there are few modern estimates. Using raw (unimputed) genotyping data, and assuming population prevalence of disease in each study population equivalent to the reported WHO prevalence rates for that country (*WHO, 2020*), we estimated polygenic heritability of susceptibility to TB in 10 contributing studies which ranged from 5 to 36% (average of 26.3%, *Supplementary file 1b*). Comparisons of the heritability estimates between studies from different geographical locations do not take into consideration the differences in environmental pressures between the included studies, and as such these estimates of heritability are only interpretable if the distribution of nongenetic determinants of TB is held constant (*Pearce, 2011*). Furthermore, variations in phenotype definition can have an impact on heritability estimates (*Supplementary file 1a*). This is supported by previous research by *McHenry et al., 2021a*, where significant differences in polygenic heritability estimates were identified between subjects with latent TB infection (LTBI), active TB, and subjects classified as resistors. (*McHenry et al., 2021a*). As this study includes data with varying methods of classifying TB cases and healthy controls (*Supplementary file 1a*), there is potential for a degree of heterogeneity and misclassification (between cases and controls) that can have an impact on the heritability estimates. Recent history has seen the near elimination of TB in several countries associated with economic development and public health action.

However, while improvement of socioeconomic standing and environment has a stronger impact than host genetics, these crude estimates of polygenic heritability do indicate that TB susceptibility is, in part, heritable. These results require future, more rigorous investigations to narrow down the level of heritable risk and pinpoint genomic loci involved by accounting for population stratification to obtain more accurate heritability estimates.

## Multi-ancestry meta-analysis identifies susceptibility loci for TB

For the primary multi-ancestry meta-analysis, MR-MEGA was used as it allows for differences in allelic effects of variants on disease risk between GWAS. Principal components (PCs), derived from a matrix of similarities in allele frequencies between GWAS, were plotted and revealed distinct separation between the three main ancestral groups included in the study (Figure 4) . To account for this, the first two PCs were included as covariates in MR-MEGA as they sufficiently accounted for the allele frequency differences between the study populations, as assessed via a QQ-plot and associated lambda inflation value (*Figure 1—figure supplement 1*, lambda = 1.00). In total, 26,620,804 variants with a minor allele frequency (MAF) > 1% and present in at least three studies were included in the analysis, of which 3,184,478 were present in all 12 datasets.

A significant association peak on chromosome 6 was identified in the *HLA class II* region (*Figure 1*). One variant (rs28383206, OR = 0.89, CI = 0.84–0.94, p-value=$8.26 \times 10^{-9}$) within this peak was associated with susceptibility to TB at genome-wide significance (p<$5.0e^{-8}$, *Figures 1–3*, *Table 2*). Both the residual heterogeneity (p-value=0.012) and ancestry-correlated heterogeneity (p-value=$5.28e^{-6}$) are significant (p-value<0.05) for the associated variant. However, the evidence of ancestry-correlated heterogeneity is much stronger than for residual heterogeneity, indicating that genetic ancestry contributes more to differences in effects sizes between GWAS than does study design (e.g., phenotyping differences and potential case–control misclassification). The association peak encompasses many *HLA-II* genes, including *HLA-DRB1/5* (major histocompatibility complex, class II, DR beta 1/5), *HLA-DQA1* (major histocompatibility complex, class II, DQ alpha 1), and *HLA-DQB3* (major histocompatibility complex, class II, DQ beta 3, *Figures 1 and 2*). While not reaching genome-wide significance, the HLA class I locus is also indirectly tagged through the association with rs2621322, in the *TAP2* (transporter 2, ATP binding cassette subfamily B member) gene, a transporter protein that restores surface expression of MHC class I molecules and has previously been implicated in TB susceptibility (*Thu et al., 2016*). *HLA-A, DQA1, DQB1, DRB1,* and *TAP2* genes have previously been linked to TB susceptibility through TB candidate gene and GWAS analysis (*Thu et al., 2016*; *Kinnear et al., 2017*; *Stein et al., 2017*; *Sveinbjornsson et al., 2016*; *Zhang et al., 2021*). The HLA-II locus encodes several proteins crucial in antigen presentation, including HLA-DR, HLA-DQ, and HLA-DP, which are widely implicated in susceptibility to infection and autoimmunity (*Kelly and Trowsdale, 2019*; *Shiina et al., 2009*).

## HLA-II

Given the strong association peak in the HLA-II locus (*Figures 1 and 2*), we imputed HLA-II alleles to fine-map this association. HLA alleles were imputed using the HIBAG R package that utilizes both genotyping array and population-specific reference panels to obtain the most accurate imputations for each individual dataset. Association testing was then conducted using an additive genetic model for each individual dataset before meta-analyzing the results (*Source data 1*, sheets 11–15).

Notwithstanding inconsistency across populations, the strongest signal in the combined global analyses is at DQA1*02:01, revealing a protective effect (OR = 0.88, 95% CI = 0.82–93, p-value=$1.3e^{-5}$, *Figure 3B*). The signal remains apparent in the six populations with the lead SNP at MAF > 2.5% and individual-level data available (p-value=0.0003). After conditioning on the lead SNP (rs28383206) in this subset, there is no residual significant association at DQA1*02:01 (p-value=0.44, *Figure 3— figure supplement 1*), suggesting that the classical allele is tagging the rs28383206 association. This observation is consistent with previous observations of HLA analysis in Icelandic (DQA1*02:01: OR = 0.82, p-value=$7.39e^{-4}$) and Han Chinese populations (DQA1*02:01: OR = 0.82, p-value=$7.39e^{-4}$), but showed opposite direction of effect in another Chinese population (DQA1*02:01: OR = 1.28, p-value=0.0193, *Figure 3B*; *Sveinbjornsson et al., 2016*; *Li et al., 2021*; *Zheng et al., 2018*).

The significant HLA associations overlap with the association peak observed in the multi-ancestry meta-analysis (*Figure 2*) but show more consistency in the direction of effects between the input studies

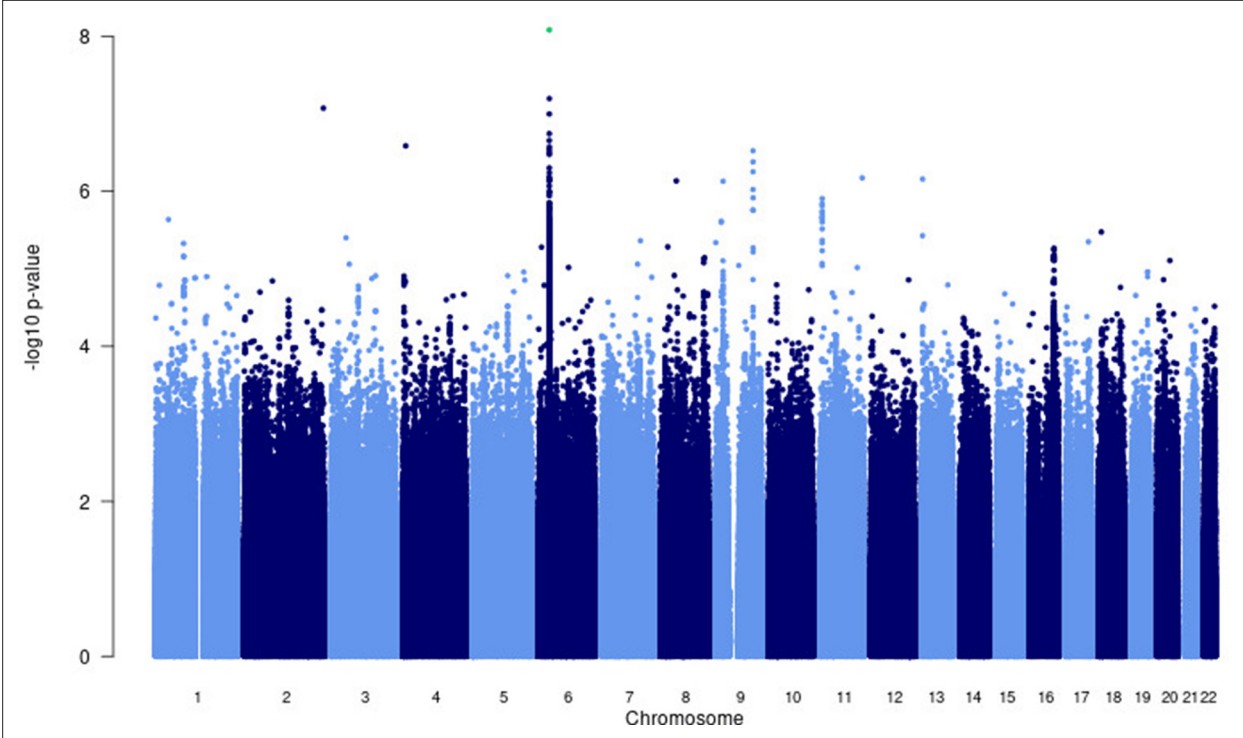

**Figure 1.** Manhattan plot of p-values (more than three studies) from the MR-MEGA analysis of all 12 datasets with genomic control reveals one significant association in the *HLA-II* region of chromosome 6 (rs28383206). Image produced using R scripts provided by MR-MEGA (*Mägi et al., 2017*), and source data file has been uploaded to https://doi.org/10.5061/dryad.6wwpzgn2s.

The online version of this article includes the following source data and figure supplement(s) for figure 1:

**Figure supplement 1.** QQ-plot (left) from the MR-MEGA analysis of all 12 datasets with genomic control correction, including two principal components (PCs) as covariates.

**Figure supplement 2.** Proportion of variants that had a significant change in association p-value (based on chi-square difference test) following the inclusion of the force of infection p-value for different p-value bins.

**Figure supplement 2—source data 1.** Proportions of variants for each p-value bin that had a significant change in association p-value, based on chi-square test for significant difference.

**Figure supplement 3.** Forest plots for the suggestive chromosome 9 peaks, rs4576509 (left) and rs6477824 (right) for the trans-ethnic MR-MEGA analysis including all 12 cohorts.

**Figure supplement 3—source data 1.** Odds ratios and 95% confidence intervals for each source data file used to plot the forest plot of the two suggestive associations rs4576509 and rs6477824.

**Figure supplement 4.** Forest plots for the suggestive chromosome 11 peak, rs12362545, for the trans-ethnic MR-MEGA analysis including all 12 cohorts.

**Figure supplement 4—source data 1.** Odds ratios and 95% confidence intervals for each source data file used to plot the forest plot of the suggestive association rs12362545.

**Figure supplement 5.** Forest plots for the suggestive chromosome 16 peak, rs35787595, for the trans-ethnic MR-MEGA analysis including all 12 cohorts.

**Figure supplement 5—source data 1.** Odds ratios and 95% confidence intervals for each source data file used to plot the forest plot of the suggestive association rs35787595.

---

compared to the lead SNPs identified in the association peak. This suggests that the rs28383206 association in the meta-analysis is tagging an HLA allele, where the different linkage disequilibrium (LD) patterns from the included ancestral populations result in the differences in effects sizes between populations at the rs28383206 association.

This variation in significant associations is, in part, attributable to the observed variation in HLA allele frequencies across all the included studies and may also reflect differential tagging of at least one unknown causal variant across populations (*Source data 1*, sheets 16–22).

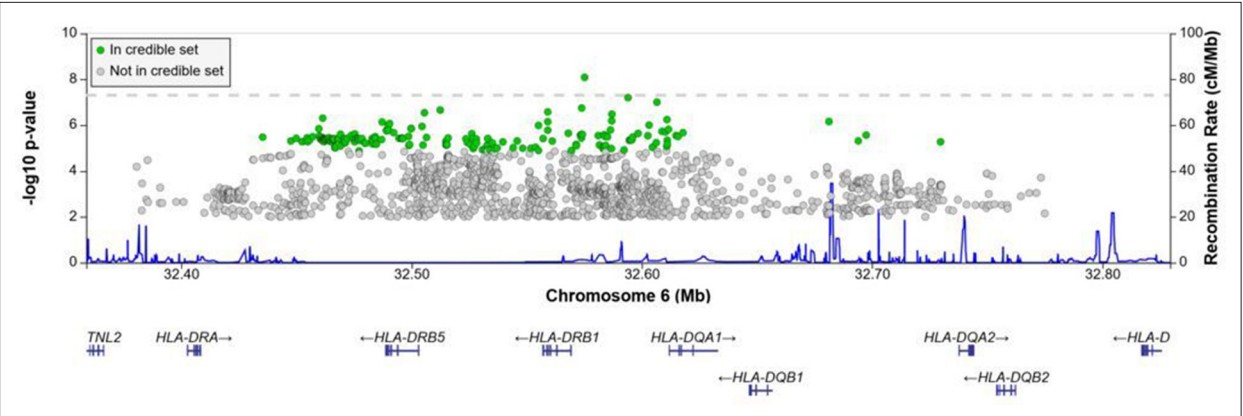

**Figure 2.** Regional association plot for the chromosome 6 *HLA-II* rs28383206 association in the multi-ancestry analysis revealing a significant peak in the HLA-II region. Image produced using the online LocusZoom database with linkage disequilibrium (LD) mapping set to 'all' and p-values>0.01 removed (*Boughton et al., 2021*), and source data file has been uploaded to https://doi.org/10.5061/dryad.6wwpzgn2s.

The variable role of classical HLA alleles in different populations could be partially due to unique infectious pressures that each geographical region faces and could also explain why different strains of *Mtb* are more or less prevalent in different regions as they adapted to the HLA profile of the population within this region. Sequencing efforts of global mycobacterial isolates find hyperconservation of class II epitopes, suggesting pathogen advantage achieved through limiting HLA-II recognition and highlighting the potential complex interplay between pathogen and host evolution in modifying class II presentation in TB infection (*Comas et al., 2010*). Previous work has shown evidence of interaction between genetic variants of the host and specific strains of *Mtb* in Ghanaian, Ugandan, South African, and Asian populations (*Möller and Kinnear, 2020*; *Müller et al., 2021*; *Correa-Macedo et al., 2019*; *Salie et al., 2014*; *Luo et al., 2015*; *Wampande et al., 2019*; *Micheni et al., 2021*; *McHenry et al., 2021b*; *McHenry et al., 2020*). These interactions provide further evidence that *Mtb* may have undergone substantial genetic evolution, in concert with host migration and evolution of different populations (*Comas et al., 2013*; *Coscolla and Gagneux, 2014*). Some studies suggest that HLA-II epitopes may have undergone regional mutations that modify HLA-II binding, and we speculate that the heterogeneity observed in HLA-II associations between regions may, at least in part, be accounted for by different pressures exerted by varying stains of *Mtb* (*Copin et al., 2016*).

## Impact of infection pressure on meta-regression

To further understand the heterogeneity across populations, we attempted to account for variation in levels of prior exposure that could serve to mask host effects given that not all controls will have been exposed to *Mtb*. In low transmission settings, more susceptible but unexposed individuals would be included as controls, who, had they been exposed to *Mtb*, might have progressed to TB disease. Overall, including each cohort's estimated prevalence of prior exposure had a significant impact on the residual heterogeneity and association statistics of 5% of the variants included in the meta-analysis (419,460/8,355,367), which at a significance level of p-value<0.05 is what is to be expected purely by chance. Separating the results into bins according to p-values revealed that the bins where the covariate had the biggest impact were for p-values in the range of $1e^{-3}$ to $1e^{-5}$ (*Figure 1—figure supplement 2*), while significant and suggestive associations reported in this study did not show any significant changes in residual heterogeneity. While the proportion of variants significantly impacted when correcting for infection pressures is low and has the biggest impact on variants with larger p-values, there was still an overall reduction in the chi-square value for the residual heterogeneity (mean chi-square value reduced by 10). This suggests that accounting for potential lifetime of infections does account for some of the observed residual heterogeneity; it is most likely not the main driving force for these residuals.

When considering the impact of force of infection, it is important to consider not only the proportion of controls ever exposed but also the impact of recurrent exposure. There is some evidence to suggest that genetic barriers to progression to TB may be overcome if the infectious dose is high (*Fox*

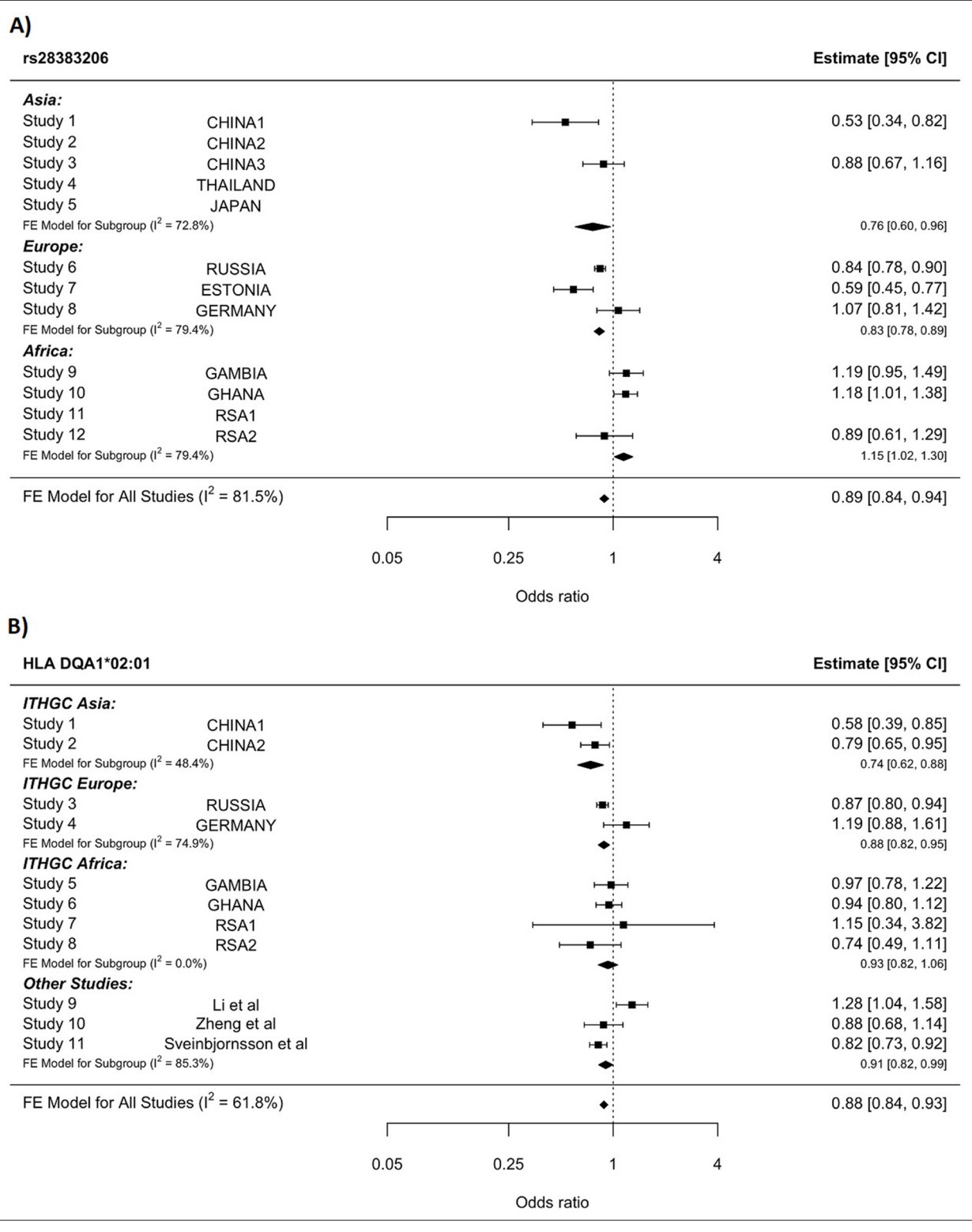

**Figure 3.** HLA conditioning analysis. (**A**) Forest plot (odds ratio and 95% confidence interval) of the significant chromosome 6 association (rs28383206) for tuberculosis (TB) susceptibility in the multi-ancestry analysis, implemented using MR-MEGA with genomic control correction (GCC). Of the 12 studies included, 8 contained this variant. Studies that did not contain the variant are included in the plot but do not have results associated with them. (**B**) Forest plot for HLA DQA1*02:01 for the eight studies included in the HLA association analysis. Other studies included were obtained from literature searches of previous studies where HLA imputation and association studies were performed (*Sveinbjornsson et al., 2016*; *Li et al., 2021*; *Zheng et al., 2018*). For source data, see *Figure 3—source data 1*.

*Figure 3 continued on next page*

*Figure 3 continued*

The online version of this article includes the following source data and figure supplement(s) for figure 3:

**Source data 1.** HLA conditioning analysis data.

**Figure supplement 1.** Results for the HLA class I and II meta-analysis of all studies overall (unconditioned) (top) and conditioned on the lead SNP for the six studies in which the lead SNP was present at minor allele frequency (MAF) > 2.5% (bottom).

**Figure supplement 1—source data 1.** Association statistics used to plot the p-value distribution for the fixed-effects meta-analysis for each HLA locus for both the conditioned and unconditioned analyses.

*et al., 1929*). Repeated exposure may be observed where TB prevalence is high, as in South Africa, and could contribute to the overall lower effects sizes observed in the GWAS enrolling RSA people. Inclusion of potential lifetime infections in meta-regression could help adjust for these effects and prove useful for not only TB, but meta-analysis of infectious diseases in general, and should be further explored.

### Other suggestive loci that did not reach significance

There were four loci with suggestive associations and strong peaks on the Manhattan plot (*Figure 1*) that did not reach significance but should still be considered as potential variants of interest (*Supplementary file 1c*). One chr9 peak (rs4576509, p-value=7.40e$^{-07}$) was intergenic (*Figure 1—figure supplement 3*) while the second (rs6477824, p-value=2.99e$^{-07}$) is located in the 5′-UTR region of the zinc finger protein 483 (*ZNF483*) gene (*Figure 1—figure supplement 3*), previously associated with age at menarche (*Demerath et al., 2013*; *Elks et al., 2010*). The chromosome 11 peak (rs12362545, p-value=1.24e$^{-06}$) is located in the PPFIA binding protein 2 (*PPFIBP2*) gene (*Figure 1—figure supplement 4*), which plays a role in axon guidance and neuronal synapse development and has previously been implicated in cancer development (*Colas et al., 2011*; *Wu et al., 2018*). The final peak (rs35787595, p-value=5.41e$^{-06}$), on chromosome 16 (*Figure 1—figure supplement 5*), is located in the craniofacial development protein 1 (*CFDP1*) gene region and involved in chromatin organization (*Messina et al., 2017*). These genes have not been previously linked to TB susceptibility and a potential role is unclear, and as a result further validation of these variants is needed before any conclusions on their impact to TB susceptibility can be drawn.

### Ancestry-specific meta-analysis

Concordance in the direction of effects of the risk allele between the ancestry-specific meta-analyses was examined to determine whether significant enrichment (above the expected 50%) exists at different p-value thresholds. Significant enrichment in the concordance of direction of effect was only observed when using the European ancestry as reference compared to the African meta-analysis results for SNPs with p-values>0.001 and <0.01 (p-value=0.0061, *Supplementary file 1d*). The lack of enrichment between the ancestries suggests significant ancestry-specific associations, which could be further compounded by the differences in local infection pressures. Due to the lack of concordance and the separation of the ancestral populations in the principal component analysis (PCA) plot (*Figure 4*), ancestry-specific meta-analysis was done.

The PCA plot (*Figure 4*) for the 12 studies (based on mean pairwise genome-wide allele frequency differences calculated by MR-MEGA) illustrates distinct separation between the three major population groups (Asia, Europe, and Africa). The separation observed between the African studies (Gambia/Ghana and RSA) is due to the high level of admixture in the RSA population. The RSA population is a five-way admixed South African population with genetic contributions from Bantu-speaking African, KhoeSan, European, and South and South East Asian populations, which explains the observed shift in the PCA plot (*Daya et al., 2013*; *Figure 4*).

**Table 2.** Significant and suggestive associations (p-value ≤1e$^{-5}$) for the multi-ancestry analysis including data from all 12 datasets implementing MR-MEGA analysis with GCC.

| Marker name | Chromosome | Position | Gene | Location | CADD score | EA | NEA | EAF | Sample size | Datasets | p-Value |
|---|---|---|---|---|---|---|---|---|---|---|---|
| rs28383206 | 6 | 32575167 | *HLA-DRB1* | Intergenic | 7.6 | G | A | 0.168 | 25,059 | 8 | 8.26e$^{-09}$ |

GCC, genomic control correction; EA, effect allele; EAF, effect allele frequency; NEA, noneffect allele.

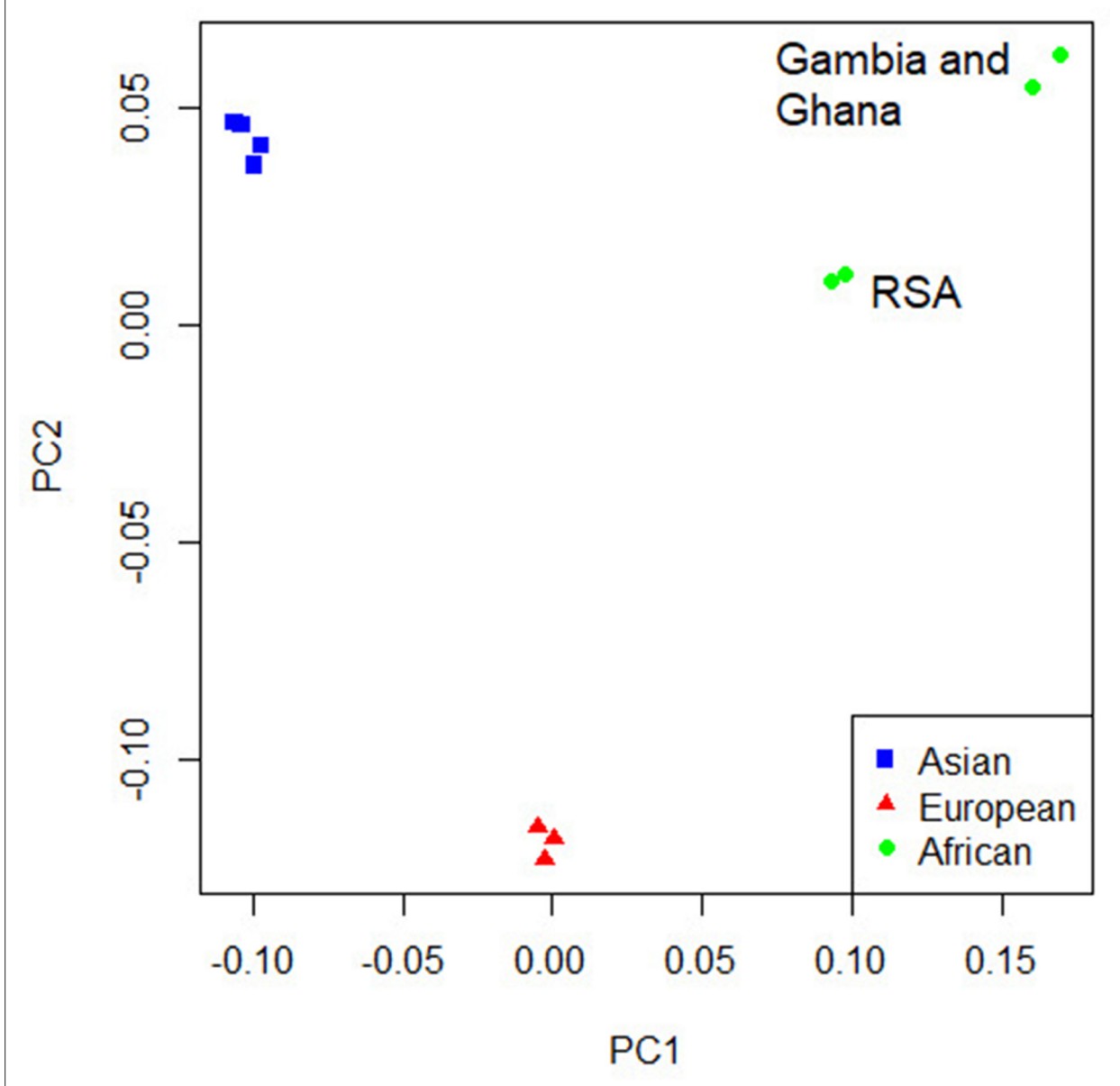

**Figure 4.** Principal component analysis (PCA) plot of all 12 studies based on the MR-MEGA mean pairwise genome-wide allele frequency differences. Image produced using the R plot function. For source data, see *Figure 4—source data 1*.

The online version of this article includes the following source data and figure supplement(s) for figure 4:

**Source data 1.** PCA source data.

**Figure supplement 1.** Manhattan plot of all p-values (≥2 studies) for the European subgroup analysis.

**Figure supplement 2.** Manhattan plot of all p-values (≥2 studies) for the Asian subgroup analysis.

**Figure supplement 3.** Manhattan plot of all p-values (≥2 studies) for the African subgroup analysis.

**Figure supplement 4.** QQ-plots for the region-specific fixed effects (FE) meta-analysis using genomic control correction (GCC) and implemented in GWAMA.

QQ-plots for the ancestry-specific analysis show no significant inflation or deflation. After removing associations without any clear peaks on the Manhattan plots (associations driven by a single study), we found no significant associations for the ancestry-specific analysis. However, suggestive peaks that did not reach genome-wide significance were identified in the European and Asian ancestry-specific analyses (*Figure 4—figure supplements 1 and 2*, *Supplementary file 1e*). Potential causes for the

lack of associations and suggestive peaks in the African analysis (*Figure 4—figure supplement 3*) are the increased genetic diversity within Africa, the inclusion of admixed samples (RSA), and the smaller sample size compared to the other ancestry-specific meta-analysis. While power can be increased through inclusion of greater genetic diversity, between-subpopulation differences in allele frequency can introduce confounding. Confounding by genetic background can result in both spurious associations and the masking of true associations. Such confounding may explain why the results observed elsewhere may not replicate in admixed samples. Removing the admixed data and analyzing only the Gambian and Ghanaian datasets also did not produce any significant results, although, clearly, the sample size was smaller.

For the European analysis (*Figure 4—figure supplement 1*), suggestive peaks were identified on chromosomes 6 (rs28383206, p-value=7.06e$^{-08}$), 8 (rs3935174, p-value=1.00e$^{-06}$), and 11 (rs12362545, p-value=1.06e$^{-07}$, *Supplementary file 1e*), while the Asian (*Figure 4—figure supplement 2*) analysis identified suggestive peaks on chromosome 6 (rs146049519, p-value**=**1.06e$^{-06}$) and 8 (rs62495207, p-value=5.10e$^{-06}$, *Supplementary file 1e*).

The suggestive peaks on chromosomes 6 and 11 in the European subgroup analysis overlap with the suggestive peaks of the multi-ancestry meta-analysis (*Figure 1*, *Figure 4—figure supplement 4*, *Supplementary file 1e*), but the suggestive peak on chromosome 8 is unique to this population (*Figure 4—figure supplement 1*, *Supplementary file 1e*). The strongest signal for this peak (rs3935174, OR = 0.87, p-value=1.00e$^{-6}$) is located in the ArfGAP with SH3 domain, ankyrin repeat, and PH domain 1 (*ASAP1*) region, which encodes an ADP-ribosylation factor (ARF) GTPase-activating protein and is potentially involved in the regulation of membrane trafficking and cytoskeleton remodeling (*Brown et al., 1998*). Variants in *ASAP1* (rs4733781 and rs10956514) have previously been linked to TB susceptibility in a TB-GWAS analysis of the same Russian population included here (*Curtis et al., 2015*). While these ASAP1 variants were present in all 12 studies and had consistent direction of effects, they presented with a strong signal in the European ancestry-specific analysis only (African and Asian p-values all ≥ 0.1). These differences in association were not driven by allele frequency differences as they are similar between the included study populations. A possible explanation for the association being observed only in the European meta-analysis is that the association is driven by the Russian dataset. rs4733781 has a strong signal in the Russian dataset (p-value=2.96e$^{-7}$), but very weak signals in all other populations included in the analysis (p-value>0.01) and is in LD with rs3935174 (r2 = 0.6935 and D′ = 0.8791) identified in our analysis. rs4733781 also did not replicate in a previous GWAS from Iceland (*Sveinbjornsson et al., 2016*), further suggesting that this association is not specific to European populations, but rather driven by the large Russian dataset included in this study.

The suggestive peak on chromosome 8 in the Asian subgroup analysis lies in an intergenic region (*Figure 4—figure supplement 2*, *Supplementary file 1e*) and the link to TB susceptibility is unclear. Finally, the suggestive region on chromosome 6 overlaps with the significant peak from the multi-ancestry analysis (*Figure 1* and *Figure 4—figure supplement 2*) and is located in the major histocompatibility complex, class II, DR beta 1 (*HLA-DRB1*), as discussed above (*Figure 4—figure supplement 2*, *Supplementary file 1e*).

## Prior associations

To determine whether associations from previously published TB-GWAS, TB candidate SNPs, and SNPs within candidate gene studies replicate in this meta-analysis, we extracted all significant and suggestive associations from prior analyses and compared these to our multi-ancestry and ancestry-specific meta-analysis results (*Luo et al., 2019*; *Schurz et al., 2018*; *Chimusa et al., 2014*; *The Wellcome Trust Case Control Consortium, 2007*; *Curtis et al., 2015*; *Mahasirimongkol et al., 2012*; *Qi et al., 2017*; *Thye et al., 2010*; *Thye et al., 2012*; *Quistrebert et al., 2021*; *Hong et al., 2017*; *Zheng et al., 2018*; *Grant et al., 2016*; *Png et al., 2012*; *Daya et al., 2014b*). In total, 44 SNPs and 36 genes were identified from the GWAS catalog, of which 33 SNPs and all candidate genes were present in our data (*Source data 1*, sheet 2). We also extracted the association statistics for a further 90 previously identified candidate genes from our multi-ancestry and population-specific meta-analysis results (*Source data 1*, sheet 2; *Naranbhai, 2016*).

Using a Bonferroni-corrected p-value of 0.0015 for the number of SNPs tested (33) as the significance threshold for replication, two candidate SNPs (rs4733781: p-value=3.22e$^{-5}$; rs10956514: p-value=0.000118; *Source data 1*, sheets 3 and 4) replicated in the multi-ancestry meta-analysis,

both located in the *ASAP1* gene region (*Curtis et al., 2015*; *Chen et al., 2019*; *Wang et al., 2018*). However, as discussed in the previous section, these associations are driven by the Russian dataset, which is the same data used by *Curtis et al., 2015*, where these associations were originally discovered (*Curtis et al., 2015*). As the Russian population included in our analysis presenting with a strong signal for these variants, there is no independent evidence for these candidate SNPs as they did not replicate in any other population.

For the Asian ancestry-specific analysis, the replicated variant was rs41553512, located in the *HLA-DRB5* gene (p-value=3.53E-05). *HLA-DRB5* is located within the HLA-II region identified in the multi-ancestry meta-analysis (*Figure 1*) and was previously identified by *Qi et al., 2017* in a Han Chinese population. The African ancestry-specific analysis did not replicate previous associations, with the lowest p-value at rs6786408 in the *FOXP1* gene (p-value=0.023). While this variant was previously identified in a North African cohort, the fact that it does not replicate here could be because of the genetic diversity within Africa and specifically the variability introduced by the five-way admixed South African population.

## Discussion

This large-scale, multi-ethnic meta-analysis of genetic susceptibility to TB, involving 14,153 cases and 19,536 controls, identified one risk locus achieving genome-wide significance, and further investigation of this region revealed significant classical HLA allele associations. This association is noteworthy given we show that there is association in other studies for the same allele (*Kinnear et al., 2017*; *Stein et al., 2017*).

Based on the significant association, rs28383206, in the HLA region identified in this multi-ancestry analysis (*Figure 3A*), HLA-specific imputation and association testing were done to fine-map the region and identify potential HLA alleles driving this association. HLA DQA1*02:01 had the strongest signal in the meta-analysis across the eight included studies (*Figure 3B*), but this signal disappeared when conditioning on the significant SNP (rs28383206). HLA DQA1*02:01 has previously been identified in an Icelandic and two Chinese populations, but the direction of effect was not consistent (*Sveinbjornsson et al., 2016*; *Li et al., 2021*; *Zheng, 2018*). Despite these inconsistencies, the association between *Mtb* and HLA class II should be explored in more detail in future studies. A study investigating the outcomes of *Mtb* exposure in individuals of African ancestry identified protective effects of HLA class II alleles for individuals resistant to TB, highlighting the importance of HLA class II and susceptibility to TB (*Dawkins et al., 2022*). HLA class II is a key determinant of the immune response in TB, and *Mtb* has the mechanisms to directly interfere with MHC class 2 antigen presentation (*Sia and Rengarajan, 2019*). This is supported by studies in mice, where mice in which the MHC class II genes were deleted died quickly when exposed to *Mtb* and died faster than the mice in which MHC class I genes were deleted (*Sia and Rengarajan, 2019*).

The p-values of residual heterogeneity in genetic effects between the studies in the multi-ancestry meta-analysis show no significant inflation between the studies. This suggests that the differences in study characteristics (phenotype definition, infection pressure, *Mtb* strain) are not the main contributor to the lack of significant associations. However, they certainly have an impact, which is further compounded with ancestry-correlated heterogeneity and other factors (e.g., socioeconomic standing). The ancestry-correlated heterogeneity p-values are generally lower than the residual heterogeneity, suggesting that genetic ancestry has a stronger impact on the differences in effects sizes between the studies. This is supported by the fact that previous TB genetic association studies have identified significant effects of ancestry on TB susceptibility (*Chimusa et al., 2014*; *Daya et al., 2014b*). However, the effects of genetic ancestry can be confounded by other factors not accounted for in this analysis, such as the differences in socioeconomic factors (including the differences in housing, employment, poverty, and access to healthcare), phenotype definitions, and differences in infection pressure between the included study populations (*Hargreaves et al., 2011*; *Duarte et al., 2018*; *Lönnroth et al., 2009*). Specifically, the lack of consistency and specificity in TB diagnosis between the included studies introduces heterogeneity and the potential for misclassification of cases and controls, which can reduce the power to detect significant associations (*Supplementary file 1a*). While this is a limitation of this study, the fact that the residual heterogeneity is overpowered by the ancestry-specific heterogeneity suggests that the phenotype definitions are not the main driver behind the lack of significant associations. For the ancestry-specific analysis, fewer studies result in there being less input

heterogeneity to account for, but the reduced sample size was not sufficient to detect any ancestry-specific genome-wide associations. This is particularly evident for the African ancestry-specific meta-analysis where the large degree of heterogeneity, which could be a result of the high genetic diversity within Africa, in combination with differences in socioeconomic factors compared to other populations included in this study, resulted in no observable suggestive association peaks (*Campbell and Tishkoff, 2008*; *Peprah et al., 2015*). Furthermore, the suggestive associations (*Supplementary file 1c and e*) reported in this study should be interpreted with care, and further validation is required before any conclusions can be drawn on the impact that they could have on TB susceptibility.

Polygenic heritability estimates revealed genetic contributions to TB susceptibility for all studies, but the level of this contribution varied greatly (5–36%), suggesting that other factors are contributing to both the lack of significant associations detected in this meta-analysis and the variation observed for the polygenic heritability estimates. These factors likely include environmental, socioeconomic, and varying levels of infection pressures, as well as genetic ancestry-specific effects between the included study populations. An individual from South Africa will face a much higher force of infection than individuals in Europe, and making the assumption that environmental circumstances are equal will significantly skew these crude heritability estimates (*Pearce, 2011*). This argument is sustained by the fact that increasing disease prevalence (higher infection pressure) increased the level of genetic contribution to TB susceptibility up to a certain point, presumably accounted for by increasingly informative control samples, after which further increasing the infection pressure will not further impact genetic susceptibility.

To determine the impact that force of infection has on the level of genetic contribution to TB susceptibility, we modeled values for proportion of people ever infected with *Mtb* to include in the multi-ancestry meta-analysis and correct for the different force of infection faced by individuals in each country. Inclusion of this covariate, however, only resulted in a significant difference for 5% of the analyzed variants, what is to be expected based on chance alone, and as such we cannot conclude that a significant portion of the observed residual heterogeneity is explained by this. Limited metadata forced us to make several assumptions about the ages of study participants and the dates on which they were enrolled. With more precise metadata, or *Mtb* infection test results in controls, the potential impact of lifetime infection could be better quantified and may contribute to elucidating genetic TB susceptibility. Multi-ancestry meta-analysis of other infectious diseases could also potentially benefit from the inclusion of force of infection covariates. It would also be important to determine whether there is a level of exposure beyond which host genetic barriers to infection are overcome (*Simmons et al., 2018*).

A single significant association was identified in this multi-ancestry meta-analysis, which is small when compared to other meta-analyses of similar size. Factors contributing to this include the difficulty in analyzing multi-ancestry data, the outdated arrays and lack of suitable reference panels for the included study populations, and heterogeneity in case and control definitions between the studies. The issue of heterogeneity in definitions is especially pronounced for this study as it included unpublished data with limited information, which does not indicate how cases were confirmed and controls were collected. The complexity of TB and generally small genetic effects suggests that larger sample sizes or alternative methods of investigation are needed. Utilizing GWAS arrays that better capture diverse populations in combination with imputation making use of larger and more diverse reference panels would allow for larger and more consistent datasets for future meta-analysis. Remapping specific areas of interest such as the *HLA*, *ASAP1,* or *TLR* using long-read sequencing would be invaluable. Increased amounts of genetic data will also allow for more accurate TB heritability analysis and permit analysis of polygenic risk scores and exploration of host–pathogen interactions.

In conclusion, this large-scale multi-ancestry TB GWAS meta-analysis revealed significant associations and shared genetic TB susceptibility architecture across multiple populations from different genetic backgrounds. The analysis shows the value of collaboration and data sharing to solve difficult problems and elucidate what determines susceptibility to complex diseases such as TB. We hope that this publication will encourage others to make their data available for future large-scale meta-analyses.

## Methods

### Data

This analysis includes 12 of the 17 published (and unpublished, *Table 1*, *Supplementary file 1*) GWAS of TB (with HIV-negative cohorts) prior to 2022 (*Schurz et al., 2018*; *Chimusa et al., 2014*; *The Wellcome Trust Case Control Consortium, 2007*; *Curtis et al., 2015*; *Mahasirimongkol et al., 2012*; *Qi et al., 2017*; *Thye et al., 2010*; *Thye et al., 2012*; *Daya et al., 2014b*). For unpublished works, we contacted researchers that were funded for genetic TB research and acquired data-sharing agreements to obtain summary statistics (or raw data) along with any metadata that was available. It excludes data from Iceland and Vietnam (*Quistrebert et al., 2021*) as they declined to share data. It excludes data from China, Korea, Peru, and Japan (*Luo et al., 2019*; *Hong et al., 2017*; *Li et al., 2021*; *Zheng, 2018*; *Sveinbjornsson et al., 2016*) as data-sharing agreements could not be finalized in time for this analysis. The Indonesian and Moroccan data were too sparsely genotyped and not suitable for reliable imputation. In addition, the Moroccan data was family-based and thus also not suitable for this meta-analysis as this would introduce confounding effects from the inclusion of related individuals (*Grant et al., 2016*; *Png et al., 2012*). Finally, cases and controls are also available within large-scale biobanks, for example, UK Biobank, which could also be leveraged in future iterations of this analysis (*Munafò et al., 2018*).

Included individuals were genotyped on a variety of genotyping arrays (*Table 1*, *Supplementary file 1*), and raw genotyping data was available for eight datasets and for the remainder association testing summary statistics were obtained to use in the meta-analysis (*Table 1*, *Supplementary file 1*). Quality control (QC) of raw genotyping data (*Table 1*, *Supplementary file 1*) was done using Plink (v1.9), followed by pre-phasing using SHAPEIT and imputation with IMPUTE2 with the 1000 genomes phase 3 reference panel (*Chang et al., 2015*; *Delaneau et al., 2013*; *Howie et al., 2009*; *Sudmant et al., 2015*). QC and imputation were done as described previously (*Schurz et al., 2018*; *Schurz et al., 2019*); briefly, we used a MAF filter of 0.025 and an individual and SNP missingness filter of 0.1. Hardy–Weinberg equilibrium threshold was set at a Bonferroni-corrected p-value according to the number of SNPs testes (0.05/number of SNPs) and samples where sex could not be determined from genotyping were also removed. Imputed data was filtered at a quality score of 0.3, prior to individual and genotype filtration steps. Prior to QC and imputation, allele orientation was corrected using Genotype Harmoniser version 1.4.15, and the genome build of all datasets was checked for consistency (GRCh37) and updated if necessary using the liftOver software from the UCSC genome browser (*Deelen et al., 2014*; *Kent et al., 2002*). The four datasets with only summary statistics available (*Table 1*, *Supplementary file 1*) were imputed and QC'd during the original investigations, but the marker names and allele orientation were checked for concordance between the summary statistics and the rest of the consortium's imputed data.

### Polygenic heritability analysis

To assess the level of genetic contribution to TB susceptibility, we estimated polygenic heritability on the individual studies for which raw genotyping data was available (*Table 1*, *Supplementary file 1*). Polygenic heritability estimates were calculated using GCTA (v1.93.2), a genomic risk prediction tool (*Yang et al., 2011*). The genetic relationship matrix was calculated for each autosomal chromosome. Raw genotype data was pruned for SNPs in LD using a 50 SNP window, sliding by 10 SNPs at a time and removing all variants with LD > 0.5. Samples were filtered by removing cryptic relatedness (--grm-cutoff 0.025) and assuming that the causal loci have similar distribution of allele frequencies as the genotyped SNPs (--grm-adj 0). Principal components were then calculated (--pca 20) to include as covariates prior to estimating heritability. Heritability estimations were transformed onto the liability scale using the GCTA software to account for the difference in the proportion of cases in the data compared to the population prevalence (*Yang et al., 2011*). The average heritability estimate was calculated by taking the mean of all estimates and the confidence intervals were estimated based on the standard error across all studies and the number of studies included.

### Meta-analysis

All variants with MAF > 1% and polymorphic in at least three studies (from at least two different ancestries) were included in the primary analysis. For the GWAS, summary statistics of each dataset variants with infinite confidence intervals were removed prior to the meta-analysis. A multi-ancestry

meta-analysis plus separate ancestry-specific analyses for Africa, Asia, and Europe were performed. MR-MEGA (Meta-Regression of Multi-Ethnic Genetic Association, v0.20), a meta-analysis tool that maximizes power and enhances fine-mapping when combining data across different ethnicities, was used for the multi-ancestry meta-analysis (*Mägi et al., 2017*). To account for the expected heterogeneity in allelic effects between populations, MR-MEGA implements a multi-ancestry meta-regression that includes covariates to represent genetic ancestry, obtained from multidimensional scaling of mean pairwise genome-wide allele frequency differences. Genomic control correction (GCC) was implemented during the MR-MEGA analysis for the individual input data (if lambda was >1.05) and output statistics, and the first two PCs, calculated from the genome-wide allele frequency differences, were included as covariates in the regression. QQ-plots of p-values and associated lambda values were used to assess the quality of results prior to downstream investigation.

For the ancestry-specific analyses, the studies were grouped by the major ancestral groups (*Table 1*, *Supplementary file 1*) and all variants with a MAF of > 1% that were observed in at least two studies were included in the meta-analysis. We performed traditional fixed-effects meta-analyses in GWAMA (v2.2.2), implementing GCC and assessed the results using QQ-plots (*Mägi and Morris, 2010*). The genome-wide significance threshold for all association testing was set at p-value=$5 \times 10^{-8}$ (*Panagiotou et al., 2012*).

## HLA imputation

To fine-map *HLA* alleles over the *HLA* locus we imputed *HLA class I and II* variants for all 8 studies for which raw data was available (*Table 1* and *Supplementary file 1*). *HLA* imputation for the *HLA class I* regions *A, B* and *C* as well as the *HLA class II* regions *DPB1, DRB1, DQB1* and *DQA1* was done using the R package HIBAG (version 1.5), implemented in the R free software environment (version 4.0.5) using the predict() command for imputation (*R Development Core Team, 2013*; *Zheng, 2018*; *Zheng et al., 2014*).

The reference datasets for HLA imputation are both genotyping panel and population-specific, and HIBAG has a database of reference data for many genotyping arrays. Each reference panel is also available for either Asian, European, or African populations or a mixture of the three (https://hibag.s3.amazonaws.com/hlares_index.html#estimates). For each dataset included for imputation, the reference panel chosen was the same as the genotyping array used for the data and the reference population was chosen to match the data as closely as possible. Asian and European reference panels were used for the Asian and European populations and African references were used for the Gambia and Ghana datasets, while mixed datasets were implemented for the admixed RSA population.

Following imputation, the HIBAG package (hlaAssocTest) command was used to implement an additive association test for the HLA alleles across the different regions limited to alleles at MAF > 2.5%. Analyses were adjusted for the first four PCs with and without the rs28383206 genotype in the model. Association testing results for the eight included studies were then combined in a fixed-effects meta-analysis using Metasoft software (*Han and Eskin, 2011*). Ancestry-specific meta-analysis grouped according to the major population groups (*Table 1*, *Supplementary file 1*) was also done using the same method.

## Estimation of infection pressure

To generate a covariate capturing the likely cumulative exposure to *Mtb* for included controls, the results of *Houben and Dodd, 2016* were adapted to produce a distance matrix to feed into the meta-analysis. The approach in this article fits a Gaussian process model of infection risk history to local data. To represent uncertainty in derived results, a sample of 200 estimated histories of the annual risk of TB infection in each country was used to calculate the expected fraction of control participants ever infected with *Mtb*, assuming that controls were uniformly aged between 35 and 44 y in 2010, which approximates the period during which controls were recruited for most of the studies. The true age of the controls was not known for all of the datasets, but as quite a substantial skew to the age distribution would be required to have an impact on the results, we believe our choice here is justified. This was done by including estimates for the potential lifetime infections for each source population as a covariate in the MR-MEGA multi-ancestry meta regression. To determine the impact of the covariate, a chi-square difference test was implemented, on an SNP-SNP basis, on the residual and association testing statistics of two meta-analysis output statistics, one including and the other

excluding the potential lifetime infections covariate (*Satorra and Bentler, 2001*). The aim was to determine whether inclusion of potential lifetime infections in the regression explained some of the residual heterogeneity.

## Concordance of direction of effect

To determine the degree to which direction of effect is shared for SNPs between the ancestry-specific meta-analysis, we followed the methodology of *Mahajan et al., 2014*. First, we identified all variants present in all 12 included datasets. Among these SNPs, we then identified an independent subset of variants in the European ancestry-specific meta-analysis showing nominal evidence of association (p-value≤0.001) and separated by at least 500 kb. The identified SNPs were then extracted from the Asian and African ancestry-specific meta-analysis results to calculate the number of SNPs that had the same direction of effect as in the European analysis. To determine whether significant excess in concordance of effect direction was present, a one-sided binomial test was implemented with the expected concordance set at 50%. This analysis was then repeated for other p-value thresholds (0.001<p≤0.01; 0.01<p≤0.5; and 0.5<p≤1), and also using the African and Asian meta-analysis results as reference.

## Acknowledgements

Computation used the Oxford Biomedical Research Computing (BMRC) facility, a joint development between the Wellcome Centre for Human Genetics and the Big Data Institute supported by Health Data Research UK and the NIHR Oxford Biomedical Research Centre. Financial support was provided by the Wellcome Trust Core Award Grant Number 203141/Z/16/Z. The views expressed are those of the author(s) and not necessarily those of the NHS, the NIHR or the Department of Health and Social Care. This work was partly supported by a Grant in-Aid for Scientific Research (B) (KAKENHI 21406006) from Japan Society for the Promotion of Science (JSPS). The clinical information and samples in Thailand, in this part, were supported by JSPS KAKENHI 17256005 and later by research grant from the Ministry of Health, Labor and Welfare (MHLW) H21-aids-12. We would like to thank all the subjects and the members of the Rotary Club of Osaka-Midosuji District 2660 Rotary International in Japan who donated their DNA for this work. We thank all members of BioBank Japan, Institute of Medical Science, The University of Tokyo, and of RIKEN Center for Genomic Medicine for their contribution to the completion of our study. This work was conducted as a part of the BioBank Japan Project that was supported by the Ministry of Education, Culture, Sports, Science and Technology of the Japanese government. As for Thai samples, we thank all of the staff and collaborators of the TB/HIV Research Project, Thailand, a research project between the Research Institute of Tuberculosis, the Japan Antituberculosis Association, and the Thai Ministry of Public Health for collecting clinical data and DNA samples. We thank the German Consortium 'TB or not TB Network' (https://www.tbornottb.de/), which was responsible for collecting the German TB samples. We acknowledge the support of the DSI-NRF Centre of Excellence for Biomedical Tuberculosis Research, South African Medical Research Council Centre for Tuberculosis Research, Division of Molecular Biology and Human Genetics, Faculty of Medicine and Health Sciences, Stellenbosch University, Cape Town, South Africa. This research was funded in whole, or in part, by the Wellcome Trust. For the purpose of open access, the author has applied a CC BY public copyright license to any Author Accepted Manuscript version arising from this submission. JJG is funded by an NIHR Academic Clinical Lectureship. APM acknowledges support from Versus Arthritis (grant reference 21754). PJD was supported by a fellowship from the UK Medical Research Council (MR/P022081/1); this UK-funded award is part of the EDCTP2 program supported by the European Union. ME was supported by an NHMRC fellowship (552496). The research was supported by the NHMRC grant 1025166. AvL and RvC are supported by the National Institute of Allergy and Infectious Diseases at NIH [R01 AI136921]. TAY is an NIHR Clinical Lecturer supported by the National Institute for Health Research. TP acknowledges funding from the National Institute for Health Research (CL-2020-21-001) and the Wellcome Trust (222098/Z/20/Z). The views expressed in this publication are those of the author(s) and not necessarily those of the NHS, the National Institute for Health Research, or the Department of Health and Social Care. AM and RM are funded by the EU project no. 2014-2020.4.01.15-0012 'Gentransmed'. BA is supported by the 'Scientific Programme Indonesia Netherlands' (SPIN) under the Royal Academy of Arts and Sciences (KNAW), the Netherlands.

# Additional information

Group author details

**International Tuberculosis Host Genetics Consortium**
**Haiko Schurz**: DSI-NRF Centre of Excellence for Biomedical Tuberculosis Research, South African Medical Research Council Centre for Tuberculosis Research, Division of Molecular Biology and Human Genetics, Faculty of Medicine and Health Sciences, Stellenbosch University,, Cape Town, South Africa; **Vivek Naranbhai**: Wellcome Centre for Human Genetics, University of Oxford, Oxford, United Kingdom; Massachusetts General Hospital, Boston, United States; Dana-Farber Cancer Institute, Boston, United States; Centre for the AIDS Programme of Research in South Africa, Durban, South Africa; Harvard Medical School, Boston, United States; **Tom A Yates**: Division of Infection and Immunity, Faculty of Medical Sciences, University College, London, United Kingdom; **James J Gilchrist**: Wellcome Centre for Human Genetics, University of Oxford, Oxford, United Kingdom; Department of Paediatrics, University of Oxford, Oxford, United Kingdom; **Tom Parks**: Wellcome Centre for Human Genetics, University of Oxford, Oxford, United Kingdom; Department of Infectious Diseases Imperial College London, London, United Kingdom; **Peter J Dodd**: Centre for Genetics and Genomics Versus Arthritis, Centre for Musculoskeletal Research, The University of Manchester, Manchester, United Kingdom; **Marlo Möller**: DSI-NRF Centre of Excellence for Biomedical Tuberculosis Research, South African Medical Research Council Centre for Tuberculosis Research, Division of Molecular Biology and Human Genetics, Faculty of Medicine and Health Sciences, Stellenbosch University,, Cape Town, South Africa; **Eileen G Hoal**: DSI-NRF Centre of Excellence for Biomedical Tuberculosis Research, South African Medical Research Council Centre for Tuberculosis Research, Division of Molecular Biology and Human Genetics, Faculty of Medicine and Health Sciences, Stellenbosch University,, Cape Town, South Africa; **Andrew P Morris**: Centre for Genetics and Genomics Versus Arthritis, Centre for Musculoskeletal Research, The University of Manchester, Manchester, United Kingdom; **Adrian VS Hill**: Wellcome Centre for Human Genetics, University of Oxford, Oxford, United Kingdom; Jenner Institute, University of Oxford, Oxford, United Kingdom; **Reinout van Crevel**: Department of Internal Medicine and Radboud Center for Infectious Diseases, Radboud University Medical Center, Nijmegen, Netherlands; Centre for Tropical Medicine and Global Health, Nuffield Department of Medicine, University of Oxford, Oxford, United Kingdom; **Arjan van Laarhoven**: Department of Internal Medicine and Radboud Center for Infectious Diseases, Radboud University Medical Center, Nijmegen, Netherlands; **Tom HM Ottenhoff**: Head Lab Dept of Infectious Diseases; Head Group Immunology and Immunogenetics of Bacterial Infectious Diseases Leiden University Medical Center, Leiden, Netherlands; **Andres Metspalu**: Estonian Genome Center, Institute of Genomics, University of Tartu, Tartu, Estonia; **Reedik Magi**: Estonian Genome Center, Institute of Genomics, University of Tartu, Tartu, Estonia; **Christian G Meyer**: Institute of Tropical Medicine, Eberhard-Karls University Tübingen, Tübingen, Germany; Faculty of Medicine, Duy Tan University, Da Nang, Viet Nam; **Magda Ellis**: Tuberculosis Research Group, Centenary Institute, Sydney, Australia; **Thorsten Thye**: School of Health and Related Research, University of Sheffield, Sheffield, United Kingdom; **Surakameth Mahasirimongkol**: Department of Medical Sciences, Ministry of Public Health, Nonthaburi, Thailand; **Ekawat Pasomsub**: Virology Laboratory, Department of Pathology, Faculty of Medicine, Ramathibodi Hospital, Mahidol University, Bangkok, Thailand; **Katsushi Tokunaga**: Genome Medical Science Project, National Center for Global Health and Medicine, Tokyo, Japan; **Yosuke Omae**: Genome Medical Science Project, National Center for Global Health and Medicine, Tokyo, Japan; **Hideki Yanai**: Fukujuji Hospital and Research Institute of Tuberculosis, Japan Anti-Tuberculosis Association, Kiyose, Japan; **Taisei Mushiroda**: RIKEN Center for Integrative Medical Sciences, Yokohama, Japan; **Michiaki Kubo**: RIKEN Center for Integrative Medical Sciences, Yokohama, Japan; **Atsushi Takahashi**: RIKEN Center for Integrative Medical Sciences, Yokohama, Japan; Laboratory for Statistical Analysis, RIKEN Center for Integrative Medical Sciences, Yokohama, Japan; Department of Genomic Medicine, Research Institute, National Cerebral and Cardiovascular Center, Suita, Japan; **Yoichiro Kamatani**: RIKEN Center for Integrative Medical Sciences, Yokohama, Japan; Laboratory for Statistical Analysis, RIKEN Center for Integrative Medical Sciences, Yokohama, Japan; **Bachti Alisjahbana**: Faculty of Medicine, Universitas Padjdjaran - Hasan Sadikin Hospital, Bandung, Indonesia; **Wei Liu**: Department of Plastic and Reconstructive Surgery, Shanghai Key Laboratory of Tissue Engineering, Shanghai Ninth People's Hospital, Shanghai Jiao Tong University – School of Medicine, Shanghai, China; National Tissue Engineering Center of China,

Shanghai, China; **A-dong Sheng**: National Clinical Research Center for Respiratory Diseases, National Key Discipline of Pediatrics, Capital Medical University, Beijing, China; Key Laboratory of Major Diseases in Children, Ministry of Education, Beijing Children's Hospital, National Center for Children's Health, Capital Medical University, Beijing, China; Beijing Key Laboratory of Pediatric Respiratory Infection Diseases, Beijing Pediatric Research Institute, Beijing, China; Children's Hospital Affiliated to Zhengzhou University, Henan Children's Hospital, Zhengzhou Children's Hospital, Zhengzhou, China; **Yurong Yang**: Ningxia Medical University, Ningxia Hui Autonomous Region, Ningxia, China

## Funding

| Funder | Grant reference number | Author |
|---|---|---|
| National Institute for Health Research | Academic Clinical Lectureship | James J Gilchrist |
| Versus Arthritis | 21754 | Andrew P Morris |
| Medical Research Council | MR/P022081/1 | Peter J Dodd |
| National Institute for Health Research | NIHR Clinical Lecturer | Tom A Yates |
| National Institute for Health Research | CL-2020-21-001 | Tom Parks |
| Wellcome | 10.35802/222098 | Tom Parks |

The funders had no role in study design, data collection and interpretation, or the decision to submit the work for publication. For the purpose of Open Access, the authors have applied a CC BY public copyright license to any Author Accepted Manuscript version arising from this submission.

## Author contributions

Haiko Schurz, Conceptualization, Data curation, Investigation, Methodology, Writing – original draft, Writing – review and editing; Vivek Naranbhai, Conceptualization, Data curation, Formal analysis, Methodology, Writing – original draft, Writing – review and editing; Tom A Yates, James J Gilchrist, Tom Parks, Peter J Dodd, Supervision, Methodology, Writing – review and editing; Marlo Möller, Eileen G Hoal, Adrian VS Hill, Resources, Supervision, Methodology, Writing – review and editing; Andrew P Morris, Software, Supervision, Methodology, Writing – review and editing; International Tuberculosis Host Genetics Consortium, Conceptualization, Data curation, Writing – review and editing

## Author ORCIDs

Haiko Schurz ![ORCID] https://orcid.org/0000-0002-0009-3409
Tom A Yates ![ORCID] http://orcid.org/0000-0002-6081-1767
James J Gilchrist ![ORCID] https://orcid.org/0000-0003-2045-6788
Marlo Möller ![ORCID] http://orcid.org/0000-0002-0805-6741
Andrew P Morris ![ORCID] http://orcid.org/0000-0002-6805-6014

## Ethics

A research collaboration agreement was signed by all contributors. Ethics approval for the meta-analysis presented here was granted by the Health Research Ethics Committee of Stellenbosch University (project registration number S17/01/013). In addition, all institutions involved in the ITHGC have ethics approval for their respective studies: China 1 and 2: The study protocol was approved by the Ethics Committee of the Beijing Chest Hospital, the 309 Hospital of the PLA, Shijiazhuang Fifth Hospital, the China PLA General Hospital, the Tongliao TB institute and the Center for Diseases Control and Prevention in Jalainuoer. China 3: Ethics approval was granted by the Ethics Committees of the Beijing Children's Hospital, the Beijing Geriatric Hospital, the Tuberculosis Hospital in Shaanxi Province, the Beijing Institute of Genomics, Chinese Academy of Sciences and the Center for Disease Control and Prevention of Jiangsu Province. Thailand: Ethics approval was granted by the Ethics Review Committee of the Ministry of Public Health in Thailand. Japan: Ethics approval was granted by the Institutional Review Board of the Center for Genomic Medicine, RIKEN Russia: Blood samples from all participants were collected and studied with written informed consent according to the Declaration of Helsinki and with approvals from the local ethics committees in Russia (St. Petersburg and

Samara) and the UK (Human Biological Resource Ethics Committee of the University of Cambridge and the National Research Ethics Service, Cambridgeshire 1 REC, 10/H0304/71). Estonia: The Estonian Bioethics and Human Research Council (EBIN) approved the Estonian Genome Center study reported in this manuscript. Germany: The study protocol was approved by the ethics committee (EC) of the University of Luebeck, Germany (reference 07-125), and was adopted by other ethics committees covering all 18 participating centres (EC of the medical faculty of the University of Goettingen; EC of the Medical Council of Hessen, Frankfurt /Main; EC of the Medical Council Hamburg; EC of the Medical Council Lower Saxony, Hannover; EC of the Medical Faculty Carl Gustav Carus, Technical University of Dresden; EC of the Medical Council Berlin; EC of the Medical Council Bavaria, Munich; EC of the Medical Faculty, Friedrich-Alexander-University Erlangen-Nuremberg; EC of the Medical Faculty of the University of Regensburg; EC of the University of Witten/ Herdecke) Gambia: Ethics approval was granted by the Medical Research Council (MRC) and the Gambian government joint ethical committee. Ghana: Ethics approval was granted by the Committee on Human Research, Publications and Ethics, School of Medical Sciences, Kwame Nkrumah University of Science and Technology, Kumasi, Ghana, and the Ethics Committee of the Ghana Health Service, Accra, Ghana. RSA A and RSA M: Ethics approval was granted by the Health Research Ethics Committee of Stellenbosch University (project registration numbers S17/01/013, NO6/07/132 and 95/072).

### Decision letter and Author response
Decision letter https://doi.org/10.7554/eLife.84394.sa1
Author response https://doi.org/10.7554/eLife.84394.sa2

---

## Additional files

### Supplementary files
• Supplementary file 1. Extended and supplementary tables. (a) Summary of ITHGC TB-GWAS datasets. (b) Polygenic heritability estimates at different TB prevalence rates. (c) Suggestive associations (p-value$\leq$1e$^{-5}$) for the multi-ancestry analysis including data from all 12 datasets implementing MR-MEGA analysis with GCC. (d) Results for the concordance in direction of effect analysis for all p-value thresholds and reference populations (for SNNP selection). (e) Suggestive associations for the European and Asian ancestry-specific FE analysis (with GCC).

• MDAR checklist

• Source data 1. Collection of data and results that are not used for figures or discussed in depth in the article but may still be valuable for other researchers working on similar topics.

### Data availability
Summary statistics of all meta-analysis will be made available on Dryad (https://doi.org/10.5061/dryad.6wwpzgn2s). The summary statistics and raw data (where available) of the individual data files cannot be made available but enquiries or requests for this data can be made through the corresponding authors or authors directly responsible for the data, listed in *Table 1*. As the ITHGC consortium has strict data transfer and sharing agreements with the original authors/owners of the data we can not ethically share the source data files in any way, be it either anonymized, de-identified or in any other form. All data that is not restricted by these data transfer and ethical agreements has been either uploaded to the online repository (https://doi.org/10.5061/dryad.6wwpzgn2s) or submitted along with this document. If any interested researchers want to apply for access to the original raw and individual GWAS datasets or any other other data currently restricted they can contact the corresponding author of this manuscript to put them in touch with the original data owners/authors, or the original data owners/authors can be contacted directly by contacting the corresponding authors listed in *Table 1*. Once the original authors/owners of the data have been contacted discussions can be had to share the data using the appropriate and ethically approved methods, which could include data transfer agreements or similar application processes.

The following previously published dataset was used:

| Author(s) | Year | Dataset title | Dataset URL | Database and Identifier |
|---|---|---|---|---|
| Schurz H, Naranbhai V, Yates TA, Gilchrist J, Parks T, Dodd P, Möller M, Hoal EG, Morris A, Hill AV | 2022 | Multi-ancestry meta-analysis of host genetic susceptibility to tuberculosis identifies shared genetic architecture | https://doi.org/10.5061/dryad.6wwpzgn2s | Dryad Digital Repository, 10.5061/dryad.6wwpzgn2s |

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
