## [Editor Report]

This article describes an important multi-ancestry meta-analysis of genome-wide association studies of susceptibility to tuberculosis. It demonstrates substantial heritability from common genetic variants, although this varies across studies. The main finding of the article is a variant in the HLA region that affects tuberculosis risk, for which the evidence is solid. The results and methods will be of interest to infectious disease researchers and human genetics researchers. The article highlights both the promise and challenges of performing multi-ancestry genetic association studies of infectious disease risk.

---

## [Decision Letter]

**Decision letter after peer review:**

Thank you for submitting your article "Multi-ancestry meta-analysis of host genetic susceptibility to tuberculosis identifies shared genetic architecture" for consideration by *eLife*. Your article has been reviewed by 3 peer reviewers, including Alexander Young as Reviewing Editor and Reviewer #1, and the evaluation has been overseen by Bavesh Kana as the Senior Editor.

Essential revisions:

The reviewers agreed that your paper is an important contribution to the effort to understand the genetics of host susceptibility to tuberculosis infection. However, some points in the paper were not clear and other points need to be expanded upon.

1) Please improve the clarity of the presentation of the results on SNP heritability. Please address reviewer #1's concern that the estimates should be transformed to the liability scale.

2) Please address reviewer #2's comments about what is driving the HLA association

3) Please address reviewers 2 and 3's comments about how your results relate to the existing associations/candidate genes discussed in the literature.

4) A more precise description of inclusion/exclusion criteria for studies in the meta-analysis is needed. It would also be better if study or ancestry specific summary statistics are released on publication as well as the main meta-analysis summary statistics.

*Reviewer #1 (Recommendations for the authors):*

The heritability estimates (as far as I can tell) are from applying GCTA to the case-control data encoded as a binary outcome. In order to make these estimates comparable across studies with different case-control ratios, the authors should transform their estimates onto the liability scale.

Why were variants showing within-ancestry heterogeneity removed?

It is hard to assess whether the test for the effect of prevalence on residual heterogeneity was well-powered enough to draw any conclusions.

The claim that there should be reduced power from inclusion of admixed samples due to increased allele frequency differences doesn't make sense to me. Greater genetic diversity should increase power (but also potentially increase confounding).

The link between finding a genome-wide significant locus in the multi-ancestry meta-analysis and the fact that tuberculosis predates the dispersal of modern humans out of Africa seems tenuous to me.

The justification for leaving UK Biobank data out of the meta-analysis doesn't seem valid. While UKB is a non-representative cohort, the case-control cohorts used in the meta-analysis are likely to be even less representative than the UKB. Why not include UKB since this could increase power substantially?

Figure S2: why is this on a different scale to the main GWAS results? Can it be put on the same scale to aid comparison with the GWAS results.

*Reviewer #2 (Recommendations for the authors):*

1. I think the paper would benefit from having a main text table with all of the nominal associations articulated. They refer to nominal associations – but no pvalues or effect sizes are provided in the main text. Since these are important findings, this should be done. They refer to Table S3 (which I cannot find).

2. I am unsure how the supplementary tables and the excel worksheets lineup. Authors refer to Figure 5, which I cannot find. Authors should carefully make sure that supplemental tables are clearly labeled and findable, along with other materials.

3. Authors present replication of ASAP1 data. Is this offering independent evidence? Is there any independent evidence of previously reported SNP associations? If not authors should say clearly in the abstract that prior known TB SNP associations failed to replicate.

Disappointing as the message is, perhaps it is one of the most important messages.

4. Authors should repeat heritability analysis with S-LDSC (using in reference LD panel) to insure robustness of GATK results. Also stratified LDSC should be used to see if there are cell-type specific annotations or gene sets that are seen consistently across the data sets. That is – it may be possible that there are clear pathways that are enriched across populations with respect to heritability captured, even if no individual alleles replicate.

5. Not clear what sort of data sharing will happen? Raw data should be share if possible – summary statistics for all of the cohorts, and of the meta-analysis.

*Reviewer #3 (Recommendations for the authors):*

First, given the heritability focus, it would be appropriate to also cite a recent paper that included heritability estimation of a number of TB phenotypes. This paper is especially relevant because it makes the point about the importance of phenotype definition to the eventual heritability estimate, a weakness that plagues some of the GWAS studies included in this paper: https://pubmed.ncbi.nlm.nih.gov/34871961/

Second, a recent paper examined HLA in an African population and did not find associations with TB. This might also be important to cite and discuss: https://pubmed.ncbi.nlm.nih.gov/35702824/

Third, a list of papers is cited about interaction between host genetic variants and strains of Mtb. Not only has this work been done in Ghana and South Africa, it has also been done in Uganda and a couple of Asian populations. This list of references really should be expanded.

Fourth, a few things in Table S1 need to be clarified. Several cells have identical phrases used for TB diagnosis ("AFB staining and culturing of Mtb from sputum samples"). Is it really "and" or is it sometimes "or" or "and/or"? Do all of these studies truly have identical definitions? Was chest x-ray ever used in the definition? This seems quite surprising given previous reviews that have detailed the phenotype definitions in some of these studies. A few cells have "NA" listed in the TB diagnosis. This must be spelt out in the footnote of the table. Also, do those papers really have no detail about the phenotype definition? There must be something.

Fifth, Table S2 presenting the polygenic heritability analyses really needs to be clarified. The column headers are not explained in the footnote. What is the difference between 0.1x', 1x', and 10x', and what does that have to do with heritability? There is also relatively little discussion of this rather complex table in the Results. It also must be clarified in the Methods whether the SNPs were thinned for LD (this is generally done in these sorts of analyses).

Sixth, the supplemental data file really should include some sort of readme to help the reviewer know what they are looking at. While the manuscript includes a list of citations of papers that were used in these candidate gene look-ups, it would be helpful if the Supplementary file included a list of genes looked-up. It also isn't clear if only GWAS hits were looked up, or well-studied candidate genes. If not the latter, they really should be included, given the wealth of research in this area.

Seventh, please clarify the infection pressure analysis. This is potentially a nice addition, but it isn't described very clearly. What are "a sample of 200 estimated histories"? There is an assumption that controls were "uniformly between 25-44 in 2010" – do the authors have age distributions for the controls in all these GWAS studies, and if the data were collected before 2010, how does that impact things?

[Editors’ note: further revisions were suggested prior to acceptance, as described below.]

Thank you for resubmitting your work entitled "Multi-ancestry meta-analysis of host genetic susceptibility to tuberculosis identifies shared genetic architecture" for further consideration by *eLife*. Your revised article has been evaluated by Bavesh Kana (Senior Editor) and a Reviewing Editor.

The manuscript has been improved but there are some remaining issues that need to be addressed, as outlined below:

Please address the remaining concerns of reviewer 2. Please also include a comment on the limitations that they may have missed including relevant datasets that they were unaware of.

*Reviewer #3 (Recommendations for the authors):*

The authors addressed most of the prior comments very thoroughly. A couple of issues remain:

1) The paper that I referenced earlier https://pubmed.ncbi.nlm.nih.gov/34871961/ also includes heritability of TB disease (see Table 3).

2) Thank you for including the thorough case and control definitions in Supplemental Table 1a. Please review it carefully – it seems that some of the case and control definitions are reversed (for example, China 1 and China 2, both have descriptions of healthy individuals under case and TB disease diagnosis under control). There are some important points seen in this table. First, TB was not always *excluded* from the control populations – often these are generic controls, and this can introduce misclassification bias. Second, the lack of specificity of TB diagnosis also introduces heterogeneity and potential misclassification as well. Not all studies used the gold standard for TB diagnosis – how does this affect interpretation?

---

## [Author Response]

Essential revisions:1) Please improve the clarity of the presentation of the results on SNP heritability. Please address reviewer #1's concern that the estimates should be transformed to the liability scale.

We have thoroughly gone over the comments from reviewer 1 and have addressed them to clarify the SNP heritability results and highlighted the fact that the heritability estimates were already transformed to the liability scale. As it was not clear that the estimates were transformed, we clarified this in the methods (page: 19) and Results section (page: 4) in the main manuscript. We have also addressed all other comments provided by reviewer #1, as detailed in this document (page: 2-4).

2) Please address reviewer #2's comments about what is driving the HLA association

We agree with the reviewers that the HLA section required more work and analysis. As a results and to address the comments of the reviewers we have completely reworked the entire HLA section to clarify the results and identify underlying factors driving the HLA association. We have also addressed all other comments and concerns brought up by the reviewer. Full responses to reviewer #2 comments are in this document (page: 4-7).

3) Please address reviewers 2 and 3's comments about how your results relate to the existing associations/candidate genes discussed in the literature.

The section on prior associations, which addresses how results relate to existing associations/candidate genes discussed in literature, has been updated to clarify how candidate genes and SNPs were selected and how they relate to the results of this meta-analysis. The exact changes to reviewer #2 are detailed in this document (page: 4-7) and responses to all comments from reviewer #3 are detailed in this document on page 7-12.

4) A more precise description of inclusion/exclusion criteria for studies in the meta-analysis is needed. It would also be better if study or ancestry specific summary statistics are released on publication as well as the main meta-analysis summary statistics.

The precise descriptions of inclusion/exclusion criteria and how unpublished studies were sought has been updated in the methods section ‘data’ of the main manuscript (page: 18) to clarify why we are limiting the analysis to the data included in this iteration of the study. Furthermore, we have updated the data sharing section of the manuscript (page: 24) to clarify which results are made available and which cannot be shared. We provide the meta-analysis output for both the global meta-analysis (which includes all studies) and the genetic ancestry specific analysis of the European, Asian, and African populations included in this study. Unfortunately, we as the consortium cannot share the summary statistics or raw data of the individual input studies as we do not have permission to do so. For access to the individual study summary statistics the original authors of the datasets need to be approached, this has been specifically mentioned in the data sharing (page: 24) section of the manuscript.

Reviewer #1 (Recommendations for the authors):The heritability estimates (as far as I can tell) are from applying GCTA to the case-control data encoded as a binary outcome. In order to make these estimates comparable across studies with different case-control ratios, the authors should transform their estimates onto the liability scale.

We thank the reviewers for pointing out that our results and methods did not convey the fact that the results were transformed onto the liability scale. For our analysis the estimate of variance explained on the observed scale was transformed to that on the underlying scale by the GCTA algorithm using linear transformation. This accounts for ascertainment bias in a case-control study, i.e., a much higher proportion of cases in the sample than in the general population. The manuscript was updated to clarify that estimates were transformed, and a footnote was added to Table S2 (now renamed supplementary file 1b) to clarify that the V(G)/Vp_L represents the transformed estimate.

The manuscript has been updated as follows on page 19.

Heritability estimations were transformed onto the liability scale using the GCTA software to account for the difference in the proportion of cases in the data compared to the population prevalence.Why were variants showing within-ancestry heterogeneity removed?

We thank the reviewers for raising this issue. Unfortunately this text had been left in from an earlier version of the manuscript but has now been remove. In earlier versions of analysis, variants showing within-ancestry heterogeneity were removed. However, using MR-MEGA software, which control for population specific effects, we did not filter based on within ancestry heterogeneity. The manuscript has been updated (page: 5) to remove this statement.

It is hard to assess whether the test for the effect of prevalence on residual heterogeneity was well-powered enough to draw any conclusions.

We appreciate the reviewer pointing this out and we agree that it is not easy to make a statement to exactly define if this test was sufficiently powered enough to draw a conclusion. For the infection prevalence analysis, we used the heterogeneity Chi-square values calculated by the MR-MEGA meta-analysis software. We used the Chi-square values from the analysis with and without the prevalence covariate added to the analysis and tested if there is a significant difference between these two Chi-square values. As this test is done on a SNP-by-SNP basis the power depends on the number of cases and controls for each SNP. As we have a fixed amount of data, we cannot do anything to change the power for this analysis without including more data. As this is the largest TB meta-analysis to data and adding more data is not possible at this point, we are unsure what we could add to clarify the conclusions from these results, and it is important to stress this aspect does not impact our main findings. Specifically in our analysis prevalence has very limited effect on the residual heterogeneity (especially compared to the heterogeneity introduced by the different ancestral background) supporting our conclusion that the background prevalence is not a major driver of factor heterogeneity.

The claim that there should be reduced power from inclusion of admixed samples due to increased allele frequency differences doesn't make sense to me. Greater genetic diversity should increase power (but also potentially increase confounding).

We thank the reviewer for pointing this out and giving us the opportunity to clarify our explanation. We agree that increased genetic diversity should increase the power, however, the inclusion of the admixed populations could introduce significant confounding effects due to differences in allele frequencies particularly between the admixed RSA and other African ancestry datasets. While the GWAS analysis of the individual RSA datasets was controlled for the effects of admixture the effects of allele frequency differences can still impact the meta-analysis. The lower sample size and reduced power of the African ancestry-specific meta-analysis compared to the other ancestry-specific meta-analysis in combination with the confounding effects can results in the lack of significant associations.

We have updated the manuscript to clarify this in page 11-12.

Potential causes for the lack of associations and suggestive peaks in the African analysis (Figure 4—figure supplement 3) are the increased genetic diversity within Africa and the inclusion of admixed samples (RSA) and the smaller sample size compared to the other ancestry-specific meta-analysis. While power can be increased through inclusion of greater genetic diversity between subpopulation differences in allele frequency can introduce confounding. Confounding by genetic background can result in both spurious associations and the masking of true associations. Such confounding may explain why results observed elsewhere may not replicate in admixed samples. Removing the admixed data and analyzing only the Gambian and Ghanaian datasets also did not produce any significant results although, clearly, the sample size was smaller.The link between finding a genome-wide significant locus in the multi-ancestry meta-analysis and the fact that tuberculosis predates the dispersal of modern humans out of Africa seems tenuous to me.

We thank the reviewer for pointing out that our statement is too strong. We agree and removed the statement from the manuscript (page: 15).

The justification for leaving UK Biobank data out of the meta-analysis doesn't seem valid. While UKB is a non-representative cohort, the case-control cohorts used in the meta-analysis are likely to be even less representative than the UKB. Why not include UKB since this could increase power substantially?

We thank the reviewer for raising this concern and agree that inclusion of more data for this manuscript would be beneficial, however, when this project and the ITHGC were established the UK Biobank data and other biobank data not included in this publication were not yet available and as such were not included. Including additional datasets at this point will require the entire body of work to be re-done and is beyond the scope of the current manuscript. We do, however, anticipate future iterations of this work including more data.

Figure S2: why is this on a different scale to the main GWAS results? Can it be put on the same scale to aid comparison with the GWAS results.

We thank the reviewer for pointing out this mistake, we have updated the figures to have the same the same scale as the other forest plots.

Reviewer #2 (Recommendations for the authors):1. I think the paper would benefit from having a main text table with all of the nominal associations articulated. They refer to nominal associations – but no pvalues or effect sizes are provided in the main text. Since these are important findings, this should be done. They refer to Table S3 (which I cannot find).

We thank the reviewer for these comments, we have updated the manuscript to include the p-values for the nominal associations throughout. We did not include a table for these nominal associations in the main manuscript as it is a long table and does not contribute significantly to the main discussion. As such we have included the table in the supplementary tables document (file name: Supplementary file 1). Table S3 (now renamed supplementary file 1c) referenced in the main manuscript can be found on page 4-5 in the file mentioned above.

2. I am unsure how the supplementary tables and the excel worksheets lineup. Authors refer to Figure 5, which I cannot find. Authors should carefully make sure that supplemental tables are clearly labeled and findable, along with other materials.

We thank the reviewer for pointing this out and agree that is vital the supplementary material is clearly integrated with the main manuscript. Accordingly, we have updated the manuscript to properly reference the supplementary tables file (Supplementary file 1) and the supplementary data excel sheet (Source data 1). The additional excel sheet contains additional information and results that are not directly discussed in detail in the main manuscript, but which could still be valuable for future research referencing this publication. We have also added a readme file (sheet 1 of the excel document) and titles to clarify the results provided in the excel sheet. Finally, the reference to Figure 5 in the main manuscript was a mistake and has been correctly updated to reference Figure 4.

3. Authors present replication of ASAP1 data. Is this offering independent evidence? Is there any independent evidence of previously reported SNP associations? If not authors should say clearly in the abstract that prior known TB SNP associations failed to replicate.

We thank the reviewer for this suggestion, and we agree that the section on ASAP1 was not clearly explained, and we have reworked this section to clarify if this association is offering independent evidence. Looking at the results the association in ASAP1 was driven by the Russian cohort as this is the only dataset where there is a strong signal for ASAP1 variants. Russia p-value for rs3935174 is 2.965610e-07 and the p-value for all other cohorts is > 0.1, except for RSA MEGA which is 0.01. Based on this we concluded that our analysis does not offer independent evidence as the Russian cohort is the same dataset in which the ASAP1 association was originally identified.

We have updated the “Ancestry-specific meta-analysis” section to clarify this (page: 13) as shown below:

“A possible explanation for the association being observed only in the European meta-analysis is that the association is driven by the Russian dataset. rs4733781 has a strong signal in the Russian dataset (p-value = 2.96e^-7^), but very weak signals in all other populations included in the analysis (p-value > 0.01) and is in LD with rs3935174 (r2=0.6935 and D’=0.8791) identified in our analysis. rs4733781 also did not replicate in a previous GWAS from Iceland ^19^, further suggesting that this association is not specific to European populations, but rather driven by the large Russian dataset included in this study.”

The “Prior associations” section was also updated (page: 13) as shown below:

“However, as discussed in the previous section, these associations are driven by the Russian dataset, which is the same data used by Curtis et al. (2015) where these associations were originally discovered^13^. As the Russian population included in our analysis presenting with a strong signal for these variants there is no independent evidence for these candidate SNPs as they did not replicate in any other population.”

Finally, we added a statement to the abstract that previous associations were not replicated in this meta-analysis (page: 2) as shown below:

“We identified one global host genetic correlate for TB at genome-wide significance (p<5e^-8^) in the human leukocyte antigen (HLA)-II region (rs28383206, p-value = 5.2e^-9^), but failed to replicate variants previously associated with TB susceptibility.”

4. Authors should repeat heritability analysis with S-LDSC (using in reference LD panel) to insure robustness of GATK results. Also stratified LDSC should be used to see if there are cell-type specific annotations or gene sets that are seen consistently across the data sets. That is – it may be possible that there are clear pathways that are enriched across populations with respect to heritability captured, even if no individual alleles replicate.

We thank the reviewers for this suggestion. However, a detailed analysis of heritability is beyond the scope of this paper. Furthermore, valid reference populations and accurate LD scores are not available for all the populations included in this publication. In future, once appropriate reference data and LD scores are available, we plan furthermore detailed analyses of heritability using the consortium dataset.

5. Not clear what sort of data sharing will happen? Raw data should be share if possible – summary statistics for all of the cohorts, and of the meta-analysis.

We thank the reviewer for pointing out that the data sharing is not clear in the manuscript. Unfortunately, we cannot share the summary statistics of the individual cohorts as this is not covered by the data sharing agreement of the consortium. Only the summary statistics for the various meta-analyses and HLA analysis are available at this point. We have updated the manuscript (page: 24) to clarify this, see below.

“Summary statistics of all meta-analysis will be made available on the Dryad online database (https://doi.org/10.5061/dryad.6wwpzgn2s). The summary statistics and raw data (where available) of the individual data files cannot be made available but enquires or request for this data can be made through the corresponding authors or authors directly responsible for the data, listed in Table 1”

Reviewer #3 (Recommendations for the authors):First, given the heritability focus, it would be appropriate to also cite a recent paper that included heritability estimation of a number of TB phenotypes. This paper is especially relevant because it makes the point about the importance of phenotype definition to the eventual heritability estimate, a weakness that plagues some of the GWAS studies included in this paper: https://pubmed.ncbi.nlm.nih.gov/34871961/

We thank the reviewer for this suggestion. However, we do not agree that the results from the suggested paper are directly comparable to our analysis, as the resister phenotype is beyond the scope of our manuscript. While we do agree that different phenotype definitions will have an impact on heritability estimates, the case definition for all our cohorts was active TB disease. While there is some variation in how exactly active TB confirmation was obtained for all our cohorts, we believe this has only a limited impact on our heritability estimates.

We updated the manuscript to mention this in page: 4, shown below.

“Furthermore, variations in phenotype definition can have an impact on heritability estimates (supplementary file 1a).”

Second, a recent paper examined HLA in an African population and did not find associations with TB. This might also be important to cite and discuss: https://pubmed.ncbi.nlm.nih.gov/35702824/

We thank the reviewer for this suggestion, however, considering the smaller sample size of the suggested publication it is difficult to relate the findings to our larger meta-analysis. Nonetheless, given our study found limited evidence of allelic associations in the available African datasets the studies the results are in fact broadly compatible. We have added a section to the discussion to highlight that the results found in the suggested study support future in depth studies of HLA class II and their role in protective effects against TB. The manuscript was updated (page: 15) as shown below:

“A study investigating outcomes of Mtb exposure in individuals of African Ancestry identified protective effects of HLA class II alleles for individuals resistant to TB, highlighting the importance of HLA class II and susceptibility to TB^62”^

Third, a list of papers is cited about interaction between host genetic variants and strains of Mtb. Not only has this work been done in Ghana and South Africa, it has also been done in Uganda and a couple of Asian populations. This list of references really should be expanded.

We thank the reviewer for pointing out these studies, the manuscript has been updated to include these references (page: 9).

“Previous work has shown evidence of interaction between genetic variants of the host and specific strains of Mtb in Ghanaian, Ugandan, South African and Asian populations^7,8,38–44^.”

Fourth, a few things in Table S1 need to be clarified. Several cells have identical phrases used for TB diagnosis ("AFB staining and culturing of Mtb from sputum samples"). Is it really "and" or is it sometimes "or" or "and/or"? Do all of these studies truly have identical definitions? Was chest x-ray ever used in the definition? This seems quite surprising given previous reviews that have detailed the phenotype definitions in some of these studies. A few cells have "NA" listed in the TB diagnosis. This must be spelt out in the footnote of the table. Also, do those papers really have no detail about the phenotype definition? There must be something.

We thank the reviewer for pointing out these issues in the table. We have updated the Table S1 (now renamed supplementary file 1a) and double checked and clarified the procedures for diagnosis of all the included datasets to include a clear phenotype definition for all included studies.

Fifth, Table S2 presenting the polygenic heritability analyses really needs to be clarified. The column headers are not explained in the footnote. What is the difference between 0.1x', 1x', and 10x', and what does that have to do with heritability? There is also relatively little discussion of this rather complex table in the Results. It also must be clarified in the Methods whether the SNPs were thinned for LD (this is generally done in these sorts of analyses).

We thank the reviewer for pointing out these issues and we have updated the manuscript to clarify the analysis and results. Table S2 (now renamed supplementary file 1b) in the supplementary data has been updated as contained results from analysis that we have discarded. The 0.1x, 1x and 10x is the prevalence multiplier where we wanted to see how the heritability estimates change with the infection pressure. This analysis was not included in the final manuscript, and the table has now been updated to remove the additional columns.

We have also updated the methods section of the heritability analysis to clarify that we used un-imputed data, and that the data was pruned for LD at a 50 SNP window, sliding by 10 SNPs at a time and removing all variants with LD greater than 0.5 (page: 19).

“The genetic relationship matrix was calculated for each autosomal chromosome (un-imputed data) which were pruned for SNPs in linkage disequilibrium (LD) using a 50 SNP window, sliding by 10 SNPs at a time and removing all variants with LD greater than 0.5.”

Sixth, the supplemental data file really should include some sort of readme to help the reviewer know what they are looking at. While the manuscript includes a list of citations of papers that were used in these candidate gene look-ups, it would be helpful if the Supplementary file included a list of genes looked-up. It also isn't clear if only GWAS hits were looked up, or well-studied candidate genes. If not the latter, they really should be included, given the wealth of research in this area.

We thank the reviewer for this feedback and suggestions. For the prior association analysis, we included previous GWAS hits as well as SNPs within well-studied candidate genes previously investigated. A list of all candidate SNPs and genes have been added to the supplementary excel data file (sheet 2), along with a readme file to explain the data in the supplementary data excel file. The readme file is also included in the excel file itself (sheet 1). The manuscript has been updated to clarify that candidate and GWAS variants were assessed. The manuscript has been updated to clarify which SNPs were included in this analysis (page: 13)

“To determine if associations from previously published TB-GWAS, TB candidate SNPs, and SNPs within candidate gene studies replicate in this meta-analysis, we extracted all significant and suggestive associations from prior analyses and compared these to our multi-ancestry and ancestry-specific meta-analysis results^6,10–18,20,23–26,31^.”

Seventh, please clarify the infection pressure analysis. This is potentially a nice addition, but it isn't described very clearly. What are "a sample of 200 estimated histories"? There is an assumption that controls were "uniformly between 25-44 in 2010" – do the authors have age distributions for the controls in all these GWAS studies, and if the data were collected before 2010, how does that impact things?

We thank the reviewer for highlighting this. The full methodology is described in Houben and Dodd. A simulation approach is used to sample historical trajectories of TB infection risk from a Gaussian process model of infection risk fitted to data, and this uncertainty is propagated through calculations. Furthermore, as we do not have age data for all the included studies, we decided to model the force of infection for 35-44 years of age. As averaging over non-uniform distributions within those age ranges is likely to give similar central estimates (perhaps smaller SD) and quite a substantial skew to the age distribution would be needed to make much of a difference to the means we propose that changes in age would not make a big impact and our chosen range is justified. Changing the year, we chose for modelling (2010) also would not have a substantial impact as TB epidemics tend to change rather slowly (~1.5% change in infection incidence per year) particularly since prevalence reflects lifetime exposure.

To help clarify this in the text without recapitulating the methods in the reference, we have changed this sentence to read (page: 21):

“The approach in this paper fits a Gaussian process model of infection risk history to local data. To represent uncertainty in derived results, a sample of 200 estimated histories of the annual risk of TB infection in each country was used to calculate the expected fraction of control participants ever infected with Mtb, assuming that controls were uniformly aged between 35-44 years in 2010, which approximates the period during which controls were recruited for most of the studies. The true age of the controls was not known for all of the datasets, but as quite a substantial skew to the age distribution would be required to have an impact on the results we believe our choice here is justified.”

[Editors’ note: what follows is the authors’ response to the second round of review.]

The manuscript has been improved but there are some remaining issues that need to be addressed, as outlined below:Please address the remaining concerns of reviewer 3. Please also include a comment on the limitations that they may have missed including relevant datasets that they were unaware of.Reviewer #3 (Recommendations for the authors):The authors addressed most of the prior comments very thoroughly. A couple of issues remain:1) The paper that I referenced earlier https://pubmed.ncbi.nlm.nih.gov/34871961/ also includes heritability of TB disease (see Table 3).

We thank the reviewer for bringing this paper to our attention. We added a reference in the Polygenic heritability section (page 4) in the previous iteration of this manuscript. We have now expanded on this to add an explanation and example from the suggested study to explain the impact that phenotype variations can have on heritability estimates and related this back to the inconsistent phenotype definitions of the studies included in this meta-analysis.

The manuscript has been updated as follows on page 4-5.

“This is supported by previous research by McHenry et al. (2021) where significant differences in polygenic heritability estimates were identified between subjects with latent TB infection (LTBI), active TB and subjects classified as resistors.^31^. As this study includes data with varying methods of classifying TB cases and healthy controls (Supplementary file 1a) there is potential for a degree of heterogeneity and misclassification (between cases and controls) that can have an impact on the heritability estimates.”

2) Thank you for including the thorough case and control definitions in Supplemental Table 1a. Please review it carefully – it seems that some of the case and control definitions are reversed (for example, China 1 and China 2, both have descriptions of healthy individuals under case and TB disease diagnosis under control). There are some important points seen in this table. First, TB was not always excluded from the control populations – often these are generic controls, and this can introduce misclassification bias. Second, the lack of specificity of TB diagnosis also introduces heterogeneity and potential misclassification as well. Not all studies used the gold standard for TB diagnosis – how does this affect interpretation?

We thank the reviewer for pointing out the fact that the case and control definitions in Supplemental Table 1a were reversed. We have addressed this and updated the table to have the correct information in the correct column. We are also grateful that the case control definitions in Supplemental Table 1a were carefully reviewed and thank the reviewer for the suggestions to highlight the impact that the heterogeneity and potential misclassification of the phenotype definitions can have. We agree that the heterogeneity and potential misclassification of the phenotype classification introduces heterogeneity and can reduce power to detect significant associations. However, we have carefully examined the results and based on the association statistics for the residual heterogeneity and ancestry-specific heterogeneity and while various factors influence the results, the ancestry-specific factors clearly have the stronger impact on the results. We reiterate this point in the discussion and again highlight the impact that the phenotype classification can have on the results in this study.

The manuscript has been updated as follows on page 16.

“Specifically, the lack of consistency and specificity in TB diagnosis between the included studies introduces heterogeneity and the potential for misclassification of cases and controls, which can reduce the power to detect significant associations (Supplementary file 1a). While this is a limitation of this study the fact that the residual heterogeneity is overpowered by the ancestry-specific heterogeneity suggests that the phenotype definitions are not the main driver behind the lack of significant associations.”